# BRIDGING THE GAP BETWEEN SEMI-SUPERVISED AND SUPERVISED CONTINUAL LEARNING VIA DATA PROGRAMMING

## ABSTRACT

Semi-supervised continual learning (SSCL) has shown its utility in learning cumulative knowledge with partially labeled data per task. However, the state-of-the-art has yet to explicitly address how to reduce the performance gap between using partially labeled data and fully labeled. In response, we propose a general-purpose SSCL framework, namely DP-SSCL, that uses data programming (DP) to pseudo-label the unlabeled data per task, and then cascades both ground-truth-labeled and pseudo-labeled data to update a downstream supervised continual learning model. The framework includes a feedback loop that brings mutual benefits: On one hand, DP-SSCL inherits guaranteed pseudo-labeling quality from DP techniques to improve continual learning, approaching the performance of using fully supervised data. On the other hand, knowledge transfer from previous tasks facilitates training of the DP pseudo-labeler, taking advantage of cumulative information via self-teaching. Experiments show that (1) DP-SSCL bridges the performance gap, approaching the final accuracy and catastrophic forgetting as using fully labeled data, (2) DP-SSCL outperforms existing SSCL approaches at low cost, by up to 25% higher final accuracy and lower catastrophic forgetting on standard benchmarks, while reducing memory overhead from 100 MB level to 1 MB level at the same time complexity, and (3) DP-SSCL is flexible, maintaining steady performance supporting plug-and-play extensions for a variety of supervised continual learning models.

## 1 INTRODUCTION

Lifelong machine learning, also known as continual learning (CL), is a machine learning paradigm that accumulates knowledge over sequential tasks (Ruvolo & Eaton, 2013a; Silver et al., 2013; Chen & Liu, 2016; Liu, 2017). It empowers machine learning at the application level such that an agent does not need to be trained from scratch with large amounts of data for every new task, as well as enabling the agent's self-improvement on previously-learned tasks by continuing to learn post-deployment. Nevertheless, researchers have identified that obtaining labeled training data is expensive (Olivier et al., 2006; Settles, 2009), which semi-supervised continual learning (SSCL) addresses(Baucum et al., 2017; Wang et al., 2021; Smith et al., 2021). As the name suggests, SSCL utilizes not only labeled data, but also leverages unlabeled task data to construct a cumulative knowledge base for learning agents, reducing labeling cost in applied machine learning.

Despite all the research efforts on SSCL, the state-of-the-art of SSCL (Baucum et al., 2017; Wang et al., 2021; Smith et al., 2021) has yet to address an elephant in the room: closing the performance gap between supervised and semi-supervised CL. Ideally, learning from $n_L$ labeled data and $n_U$ unlabeled data per task should provide the same lifelong performance as if all the $n_L + n_U$ data are labeled, but state-of-the-art SSCL frameworks have not approached this goal, and rarely consider computational cost required to do so. Moreover, multiple supervised CL tools have matured Lee et al. (2019); Yoon et al. (2018); Bulat et al. (2020) and would likely benefit by extending them to the semi-supervised setting, but current SSCL approaches are architecture-specific and such extension is non-trivial.

Motivated by the challenges above, we propose data programming (DP) (Ratner et al., 2016b) as a solution. DP is an automatic psuedo-labeling approach that collectively generates labels from noisy labeling functions, with some methods providing probabilistic guarantees on pseudo-label accuracy (Ratner et al., 2016a; Varma & Ré, 2016). Ideally, the more diverse noisy labelers are sampled, the higher quality pseudo-labels can be produced - approaching perfect labeling accuracy. Therefore, upon every task in SSCL, by training a pseudo-labeler via DP and then cascading both ground-truth-labeled and pseudo-labeled data into a supervised CL model, we are able to approach high quality pseudo-labeling and decrease the performance gap between semi-supervised and supervised CL. This procedure is also benefited from the small overhead of DP in terms of both time and memory, lowering resource costs on large amounts of unlabeled data and long task sequences. Furthermore, the cumulative knowledge along CL can assist the pseudo-labeler performance, leveraging transfer-ability analysis metrics (Nguyen et al., 2020; Tan et al., 2021; Pandy et al., 2022; Tran et al., 2019). Intuitively, the more similar two tasks, the more similar ways they should handle the unlabeled data to shrink the gap. In practice, noisy labeling functions from previous tasks can be retained and transferred to new tasks based on task transferability, utilizing cumulative knowledge to self-teach the pseudo-labeler throughout the lifelong sequence. This framework design also allows supervised CL approaches to be extended in a plug-and-play fashion, by decoupling the pseudo-labeling and continual learning modules.

Experiments on standard image classification benchmarks show DP-SSCL achieves final accuracy and catastrophic forgetting comparable to supervised CL on fully labeled data. Moreover, DP-SSCL outperforms existing SSCL tools with up to $25\%$ higher final accuracy and lower catastrophic forgetting, while reducing the memory overhead for unlabeled data processing from the $100$ MB to the $1$ MB level with the same time complexity. Additionally, ablation studies show DP-SSCL maintains steady continual performance at increasing sizes of unlabeled data per tasks, over longer task sequences, and using different knowledge transfer mechanisms.

## 2 BACKGROUND AND RELATED WORK

### 2.1 LIFELONG LEARNING/CONTINUAL LEARNING (CL)

The primary goal of continual or lifelong learning is to learn tasks consecutively, exploiting forward transfer to facilitate the learning of new tasks while retaining performance on previous tasks without catastrophic forgetting. The vast majority of research focuses on supervised methods, using techniques such as weight importance vectors (Fernando et al., 2017; Aljundi et al., 2019) to cache critical pathways and prevent catastrophic forgetting, factorized transfer to decompose the model parameter space (Ruvolo & Eaton, 2013b; Bulat et al., 2020; Lee et al., 2019), deconflicting projections to ensure that new tasks are trained using unused capacity within the deep network (Farajtabar et al., 2019; Zeng et al., 2019; Saha et al., 2021), and dynamically expanding networks that grow to accommodate tasks (Veniat et al., 2021).

### 2.2 SEMI-SUPERVISED CONTINUAL LEARNING (SSCL)

Recently, techniques have been developed for CL in semi-supervised settings to take advantage of unlabeled data.

A common procedure of SSCL is to pseudo-label these unlabeled data for training set augmentation. For instance, CNNL (Baucum et al., 2017) fine-tunes a lifelong learning model by repeatedly pseudo-labeling unlabeled data using the model itself, and then augments its training set with the newly-labeled data. Alternatively, DistillMatch (Smith et al., 2021) identifies unlabeled data points that are possibly seen in previous tasks by an out-of-distribution detector, and pseudo-labels them using distilled accumulated knowledge. A third example is ORDisCo (Wang et al., 2021), which trains a GAN-based pseudo-labeler in parallel with a lifelong learning model by using a three-branch network, which enables it to learn the joint distribution of data and labels simultaneously. Similarly, Semi-ACGAN (Brahma et al., 2021) utilizes GAN for training task-dependent classifiers while using the unlabeled data only to train the discriminator of GAN for the source of data (real vs fake). The last example (Ho et al., 2022) combines prototypical learning for pseudo-labeling with meta learning to achieve both the label generation on the unlabeled data and fast adaptation to any task in the continual learning scenario. Under pseudo-labeling, bridging the gap towards supervised CL

becomes simple: the higher pseudo-labeling accuracy, the closer performance to CL on fully labeled data. Straightforwardly, to shrink the gap is to improve the pseudo-labeler.

The methods above all generate labels for the unlabeled data by using the classifiers that are trained for task objectives, and their novelty comes from supplementary design decisions that tackle issues of SSCL such as catastrophic forgetting and pseudo-label consistency. A major shortcoming of these approaches arises due to the frequently observed tendency of neural network models to have overconfidence in their predictions (Guo et al., 2017a; Nguyen et al., 2015; Hein et al., 2019); as a result neural-net-based labelers may confidently generate incorrect pseudo-labels, especially when encountering novel tasks or out-of-distribution data. On the other hand, our DP-SSCL method utilizes DP, which provides theoretical guarantees on the quality of labels, making the pseudo-labeling process more robust to new data from the novel tasks that arise in CL settings. Moreover, these existing SSCL tools require large additional memory and computation overhead for pseudo-labeling, such as storage of a ResNet-34 backbone (Smith et al., 2021), GAN networks (Wang et al., 2021; Brahma et al., 2021) or MAML (Ho et al., 2022), while DP-SSCL minimizes this overhead by storing light-weight labeling functions facilitated by data programming, which we describe below.

Additionally, techniques that do not depend directly on pseudo-labels have been proposed for stable semi-supervised continual learning. For instance, pseudo-gradient learners (Luo et al., 2022) are trained instead of pseudo-labelers to provide auxiliary gradients for the model update from the unlabeled data. This is useful for the case that the unlabeled data may include instances of unknown classes. The other work, CCIC (Smith et al., 2021), adapts MixMatch to use pseudo-labels of the unlabeled data as consistency regularization target rather than training target. Still, these methods avoid the usage of pseudo-labels as direct training target, but lack theoretical guarantees on the benefit of these approaches as provided by our DP-SSCL.

## 2.3 DATA PROGRAMMING (DP)

Data programming (DP) is an approach to automatically produce pseudo-labels on an unlabeled data set $X_U$, given a labeled data set $(X_L, y_L)$ (Ratner et al., 2016b). The idea is to ensemble a set of noisy weak labeling functions (WLFs) with each performing only slightly better than random guessing on their own, such that the ensembled pseudo-labels, or strong labels, achieve high accuracy. Therefore, DP papers generally discuss two problems in cascading order: (1) how to generate these WLFs and (2) how to do the ensembling.

To address the upstream problem on WLF generation, one simple way is to collect manually designed functions by experts (Ratner et al., 2016b). One succeeding tool, namely Snuba (Varma & Ré, 2016), iteratively trains multiple sets of WLFs, and then from every set selects the top $k$ WLFs ranked by a score

$$s = w * F1(y_L, \hat{y}_L) + (1 - w) * Jaccard(\hat{y}_U) \tag{1}$$

where $\hat{y}_L$ and $\hat{y}_U$ are the predicted label vectors by a WLF on the labeled and unlabeled data respectively. The score consists of a performance metric (F1) on the labeled set, as well as a diversity metric (Jaccard distance (Jaccard, 1902)), with a weighting factor $w$ usually $= 0.5$. The selected WLFs that pass this pruning step form the committed labeler set $F$.

Then, for the downstream problem on ensembling, multiple techniques can be applied. For example, majority voting is one of the brute force methods. Other existing techniques such as repeated labeling could also apply (Ipeirotis et al., 2014). Among these methods, Snorkel (Ratner et al., 2016a) learns a generative model on top of committed WLFs in form of

$$\pi_\phi(\hat{Y}_U, Y_U) = \frac{1}{Z_\phi} \exp(\phi^T \hat{Y}_U Y_U) \tag{2}$$

where $\hat{Y}_U$ and $Y_U$ are the aggregated label matrices of all committed WLF labels and ground-truth labels, $\phi$ is the parameters and $Z_\phi$ is a normalization factor. Snorkel trains this generative model such that it labels $y_U$ with high accuracy. This ensembling requires each WLF to have higher accuracy than random guessing, which is a low requirement for learners.

When the downstream ensembling uses a generative model as in equation 2, Snuba provides a probabilistic guarantee that the accuracy of the generative model on labeled data and unlabeled data has a maximum difference of $\epsilon$ with probability $1 - \delta$. This guarantee exists because Snuba checks an exit

condition on WLFs, such that each committed WLF before termination must have a certain level of confidence on $d$ data points, with

$$d \geq \frac{1}{2(\gamma - \epsilon)^2} \log \left( \frac{2|F|^2}{\delta} \right). \tag{3}$$

Here, $\gamma$ is the measured error, i.e., the difference of accuracy between the generative model and WLFs on the labeled data. Please refer to the original paper for a more detailed proof. We also extend the proof of this guarantee to the continual learning setting in Appendix A.

## 2.4 TRANSFERABILITY

In transfer learning, transferring a trained model to a target task that shares no common features with the source task typically hurts the performance of the model, even worse than learning only the target task. As such, understanding the similarity of tasks and measuring transferability of a learning model from one task to another is a key aspect of not only transfer learning but also continual learning. An intuitive metric of transferability is the accuracy of the source model on the target data (Tran et al., 2019; Dhillon et al., 2020): measuring how well the trained model performs on the target task. LEEP (Nguyen et al., 2020) extends this metric by weighting the likelihood of the model with empirical conditional distribution of the target label given the source label. OTCE (Tan et al., 2021) quantifies the distance between two classification tasks as a sum of domain difference and task difference. The domain difference – the difference of the data distribution – is computed by optimal transport theory with entropic regularization, and the task difference – the difference of the classification objectives such as a set of classes – is derived from conditional entropy using the optimal coupling matrix of the optimal transport problem. GBC (Pandy et al., 2022) adopts the Bhattacharyya coefficient that measures the amount of overlap between two distributions in the feature space of the source model. Based on the positive results by using transferability score in transfer learning, this score can help figuring out the reusable knowledge of earlier tasks more effectively.

## 3 PROBLEM FORMULATION

We aim to solve the following problem: How to design an SSCL framework, such that it is able to (1) minimize the performance gap between using partially labeled data and fully labeled data per task at low cost, (2) allow cumulative knowledge transfer to assist handling of unlabeled data, and (3) extend from arbitrary existing supervised CL frameworks?

Formally, in an SSCL problem, a lifelong learner will face a sequence of classification tasks $\{\mathcal{Z}^{(1)}, \mathcal{Z}^{(2)}, \ldots\}$, with each task $\mathcal{Z}^{(i)}$ having data space $\mathcal{X}^{(i)} \subseteq \mathbb{R}^{d_i}$, label space $\mathcal{Y}^{(i)} = \{1, \ldots, c_i\}$, and a joint distribution $\mathcal{D}^{(i)} : \mathcal{X}^{(i)} \times \mathcal{Y}^{(i)} \mapsto [0,1]$ governing the data-label pairs. For task $\mathcal{Z}^{(i)}$, we are given $n_L^{(i)}$ labeled training data $(X_L^{(i)}, y_L^{(i)}) \sim \mathcal{D}^{(i)}$, $n_U^{(i)}$ unlabeled training data $X_U^{(i)} \sim \mathcal{X}^{(i)}$ and $n_T^{(i)}$ testing data $(X_T^{(i)}, y_T^{(i)}) \sim \mathcal{D}^{(i)}$.

From this setting, we consider a two-part SSCL framework consisting of an upstream pseudo-labeler and a downstream supervised CL module that are isolated from each other's processes. Specifically, at task $\mathcal{Z}^{(i)}$, a pseudo-labeling function $\pi^{(i)} : \mathcal{X}^{(i)} \mapsto \mathcal{Y}^{(i)}$ is first trained. Then, the unlabeled data $X_U^{(i)}$ will be labeled with $\hat{y}_U^{(i)}$ by this function. Next, labeled data $(X_L^{(i)}, y_L^{(i)})$ and pseudo-labeled data $(X_U^{(i)}, \hat{y}_U^{(i)})$ will be given to a supervised CL module for continual learning. With this design, users are able to plug-and-play existing well-developed supervised CL tools and augment them into SSCL, addressing sub-problem (3).

The design of the pseudo-labeling procedure addresses sub-problems (1) and (2). Intuitively, the higher the pseudo-label quality, the smaller the gap is between using partially labeled and fully labeled data (as if each task $\mathcal{Z}^{(i)}$ has all $n_L^{(i)} + n_U^{(i)}$ labeled). Hence, (1) asks for a labeler that can ideally approach perfect labeling by learning every underlying distribution $\mathcal{D}^{(i)}$, preferably with guaranteed label quality, and low time and memory overhead. Furthermore, (2) asks for a cumulative knowledge base and its input/output algorithm specifically for the pseudo-labeler. In the next section, we provide a concrete framework that meets these requirements.

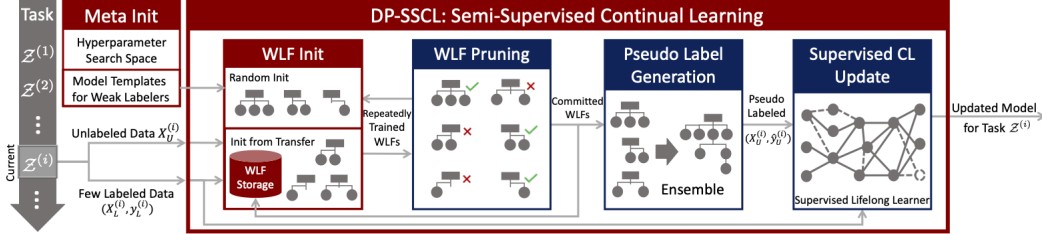

Figure 1: The DP-SSCL framework, our contribution marked as red and adopted modules blue.

# 4 THE DP-SSCL FRAMEWORK FOR SEMI-SUPERVISED CONTINUAL LEARNING

In this section, we introduce our proposed framework, Data Programming-based SSCL, or DP-SSCL. The overall workflow is illustrated in Figure 1. As shown, after a meta initialization, upon every new task, four modules are called in cascading order: (1) WLF initialization and training, (2) WLF pruning, (3) pseudo-label ensembling and (4) supervised CL model updating. Our framework also includes light-weight storage for all committed WLFs throughout all the tasks.

## 4.1 META INITIALIZATION

In a typical CL setting, the tasks are not available at once, but we are able to assume some general knowledge for the type of tasks (Liu, 2017; Ruvolo & Eaton, 2013a; Silver et al., 2013). For example, what input dimensions and what latent space that encodes representative features. Therefore, we perform a meta initialization step to allow users to input these prior information before the first task. Such prior knowledge includes a training hyperparameter search space, in which hyperparameters such as learning rate, batch sizes and number of epochs for WLF training will be grid-searched later as new tasks arise. Other prior knowledge includes model templates, which define the architectures for WLFs, which are typically decision trees, regressors, or small neural networks not exceeding 5 layers or 1000 parameters. The WLF storage is initialized to be empty.

## 4.2 WLF INITIALIZATION VIA KNOWLEDGE TRANSFER

Upon the arrival of a new task, DP-SSCL first acquires a set of WLFs, each in form of $f_{weak}$ : $\mathcal{X}^{(i)} \mapsto \mathcal{Y}^{(i)} \cup \{0\}$, where an additional 0 label means the confidence of $f_{weak}$ on a data point is lower than a given threshold $t$ such that labeling is abstained. To produce WLFs that can ensemble to an accurate pseudo-labeler, we consider two sources of WLF initialization: (1) random sampling on the pre-defined model templates in meta initialization, leveraging the user's prior knowledge of the task and (2) transfer from previous tasks, leveraging knowledge accumulated throughout the continual learning process.

Knowledge is represented in the form of previously committed WLFs, that is, WLFs that enters the pseudo-labeler ensembling procedure in previous tasks $\mathcal{Z}^{(1)}, \ldots, \mathcal{Z}^{(i-1)}$. Upon task $\mathcal{Z}^{(i)}$, DP-SSCL evaluates the suitability of the previously committed WLFs for transfer to the current task using transferability measures, such as the LEEP or OTCE transferability score introduced in Section 2.4, and selects from the cached WLFs using the procedure shown in Algorithm 1.

Algorithm 1 returns one of the previously committed WLFs to initialize a new weak labeling function. In this algorithm, the suitability score and the committed WLF selection criteria can be designed based on prior data or task knowledge. For example, one can use only the task transferability score to sample tasks relevant to the current task, and then select a weak labeling function at random by treating WLFs of the sampled task uniformly. On the other hand, it is also possible to consider the labeling accuracy of the earlier WLFs in addition to task transferability for the selection of WLFs. Since there is no guarantee that all previously encountered tasks are closely related to the current task, a suitability score threshold $\phi$ excludes negatively correlated tasks from the pool of WLFs for transfer. Similarly, we introduce another parameter – probability of initializing a WLF by transfer $\rho$

---

**Algorithm 1** WLF Transfer

---

**Input**: WLF storage $\mathcal{W}$, CL model $\mathcal{M}$, and the current task data $(X_L^{(i)}, y_L^{(i)})$
**Parameter**: Suitability score threshold $\phi$
**Output**: A selected WLF $\tilde{f}_{weak}$ in the storage $\mathcal{W}$

1: $s_{WLF} \leftarrow$ computeSuitabilityScore($\mathcal{W}, \mathcal{M}, X_L^{(i)}, y_L^{(i)}$)
2: $p_{WLF} \leftarrow$ convertSelectionProbability($s_{WLF}, \phi$)
3: $\tilde{f}_{weak} \leftarrow$ randomSelection($\mathcal{W}, p_{WLF}$)
4: **return** $\tilde{f}_{weak}$

---

– to control the ratio of the transferred WLFs to newly generated WLFs for the current task, which allows for the some WLFs to be initialized from scratch to maintain diversity in the WLF pool.

### 4.3 WLF Training and Pruning

After initialization, the framework obtains a set of both transferred and randomly generated WLFs, which are fine-tuned by labeled data at current task. The training hyperparameters such as learning rate, batch size and epochs are selected from within the bounds specified during the meta initialization phase. The trained WLFs then pass through a Snuba pruner (Varma & Ré, 2016), which commits only the top functions ranked by score computed by Equation equation 1. To improve the diversity of each WLF, we adopt bootstrapping on the training data, such that each WLF is trained on a randomly selected subset of the labeled data, with the boostrapped size specified during meta initialization. The initialization and training procedure is repeated until either a maximum size of committed WLFs is met, or if the condition in equation 3 will become violated in the next iteration. Consequently, although we include transferred functions in this procedure, our framework still maintains the Snuba guarantee on pseudo-labeler quality. A detailed proof of how DP-SSCL maintains this guarantee in the continual learning setting is presented in Appendix A.

### 4.4 Pseudo-labeler Ensembling and Continual Model Update

The committed WLFs of task $\mathcal{Z}^{(i)}$ enter ensembling, where different aggregators can be used. For instance, majority voting is a simple method to combine WLFs, as well as repeated labeling (Ipeirotis et al., 2014). More advanced ensembling methods are available, such as training a Snorkel (Ratner et al., 2016a) generative model $\pi^{(i)}$ in the form of equation 2. We empirically evlauate different ensembling techniques in an ablation study in Section 5.2. Via the ensembled labeler, the framework obtains pseudo-labels $\hat{y}_U^{(i)}$. After ensembling, if the confidence of the pseudo-labels are available, the pseudo-labels can be further adjusted by confidence calibration (Platt, 1999; Guo et al., 2017b).

Last, together with labeled data $(X_L^{(i)}, y_L^{(i)})$, the pseudo-labeled data $(X_U^{(i)}, \hat{y}_U^{(i)})$ enters a supervised continual learning module to update its model to learn a function in the form of $f : \bigcup_{j=1}^{i} \mathcal{X}^{(j)} \mapsto \bigcup_{j=1}^{i} \mathcal{Y}^{(j)}$. For the underlying model, our framework supports many state-of-the-art continual learning approaches, such as DF-CNN (Lee et al., 2019), TF (Bulat et al., 2020), and DEN (Yoon et al., 2018), which we demonstrate below.

## 5 Experimental Evaluation

In this section, we first explain our instantiation of DP-SSCL from ablation study results on ensembling methods and WLF transfer. Then, we explore how DP-SSCL's lifelong performance with partially labeled data compares to that of supervised CL using fully labeled data sets, as well as how DP-SSCL performs compared to existing SSCL methods. More experiments and discussion are presented in Appendix D.

Table 1: Comparison of different pseudo-labeler ensembling methods.

| Continual Learner | Labeling Method | MNIST Final Accuracy | | |
|---|---|---|---|---|
| | | $n_U^{(i)} = 60$ | $n_U^{(i)} = 120$ | $n_U^{(i)} = 240$ |
| | Majority Voting | $96.5_{\pm 0.5}$ | $89.1_{\pm 12.6}$ | $94.8_{\pm 0.3}$ |
| DP-SSCL (TF) | Repeated Labeling | $89.7_{\pm 1.1}$ | $84.7_{\pm 0.7}$ | $77.8_{\pm 0.3}$ |
| | Snorkel | $\mathbf{97.0}_{\pm \mathbf{0.5}}$ | $\mathbf{97.0}_{\pm \mathbf{0.2}}$ | $\mathbf{96.5}_{\pm \mathbf{0.2}}$ |
| | Majority Voting | $96.0_{\pm 0.3}$ | $95.6_{\pm 0.3}$ | $93.3_{\pm 1.0}$ |
| DP-SSCL (DF-CNN) | Repeated Labeling | $91.8_{\pm 0.5}$ | $86.1_{\pm 1.0}$ | $77.4_{\pm 0.6}$ |
| | Snorkel | $\mathbf{96.5}_{\pm \mathbf{0.3}}$ | $\mathbf{96.4}_{\pm \mathbf{0.3}}$ | $\mathbf{95.2}_{\pm \mathbf{0.8}}$ |

## 5.1 SETUP

We show that DP-SSCL is a general framework for SSCL by using different combinations of algorithms for strong label generation (majority voting, repeated labeling, and Snorkel) and supervised CL (TF Bulat et al. (2020), DEN Yoon et al. (2018), and DF-CNN Lee et al. (2019)). Our experiments employ continual learning versions of standard image classification benchmarks, including MNIST LeCun & Cortes (2010), CIFAR-10, and CIFAR-100 Krizhevsky (2009). For each task, we hold out 10% of labeled and unlabeled data for validation. We evaluate performance using (1) peak per-task performance metrics, (2) final task performance metrics, and (3) forgetting metrics, with forgetting metrics measured as backward-transfer (Ruvolo & Eaton, 2013b; Lopez-Paz & Ranzato, 2017). The detailed experimental setup , including hyperparameter selection and evaluation metrics, is explained in the following subsections as well as in Appendix B.

## 5.2 ABLATION STUDIES

The following two ablation studies are evaluated on validation data to select components for the main experiments in Section 5.3, which is then evaluated on separate testing data. More ablation studies are detailed in Appendix D.

**Ensembling Methods** We evaluate three methods of pseudo-labeler ensembling discussed in Section 4.4: majority voting, repeated labeling, and Snorkel. As shown in Table 1 (refer Table 7 in Appendix D for details), we see that Snorkel obtains consistently strong performance across continual learning algorithms, and amounts of unlabeled training data. Consequently, we use Snorkel as the label generator within DP-SSCL for all other experiments.

**WLF Transfer** We instantiate our framework with knowledge transfer in WLFs by using DF-CNN (CL model), LEEP score (suitability score) and binary CIFAR10 experiment. As depicted in Figure 2, both peak per-task accuracy and final accuracy slightly increase in comparison to DP-SSCL without WLF transfer (black dashed lines) when approximately a half of WLFs are initialized via transfer. Due to the fact that the increment is small, we pick a high LEEP score threshold that results in no transfer for all the remaining experiments, without a loss in generality. Though the accuracies are within 95% confidence interval of the baseline, these enhanced CL performances show its positive potential of knowledge transfer in weak labeling functions.

## 5.3 MAIN RESULTS

### 5.3.1 SSCL PERFORMANCE

Based on the instantiation of Snorkel ensembler and no LEEP transfer, we compare DP-SSCL to three state-of-the-art SSCL algorithms: CNNL (Baucum et al., 2017), ORDisCo (Wang et al., 2021), and DistillMatch (Smith et al., 2021). To enable fair comparisons, we replicate the experimental conditions and compare our approach to the best results reported in each algorithm's original publication. Specifically, we replicate the instance-incremental learning experiments of CNNL on MNIST and CIFAR-10, and the class-incremental learning experiments of ORDisCo and DistillMatch on CIFAR-10 and CIFAR-100, respectively. The replicated experimental protocols are summarized below; please refer to the original papers for detailed setups.

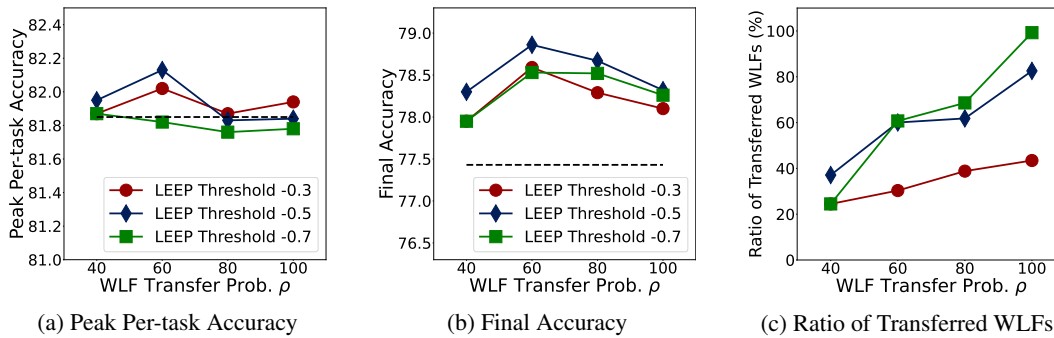

Figure 2: SSCL performance when transferring WLFs of earlier tasks.

Table 2: Instance-incremental SSCL results.

| | MNIST | | CIFAR-10 | |
|---|---|---|---|---|
| Approach | Final Accuracy | Batches to Saturation | Final Accuracy | Batches to Saturation |
| CNNL | **90.0** | 26 | 45.7 | **25** |
| DP-SSCL Labeled | $90.0_{\pm 0.4}$ | $3.3_{\pm 1.6}$ | $54.2_{\pm 0.5}$ | $26.5_{\pm 1.1}$ |
| *Fully Labeled* | $99.0_{\pm 0.1}$ | $17.0_{\pm 4.0}$ | $57.6_{\pm 0.5}$ | $27.1_{\pm 1.7}$ |

For instance-incremental experiments, all tasks are present at every epoch, but subsequent epochs contain different batches of unlabeled data. Each experiment was run over 10 random seeds. As shown in Table 2, DP-SSCL achieves 90% of final testing accuracy as if fully labeled data is used, comparable to CNNL in MNIST and higher in CIFAR-10. Moreover, on both benchmarks DP-SSCL has comparable or higher sample efficiency measured in batches to saturation, which means the first epoch where a 3-batch sliding window average meets or exceeds the final accuracy.

For class-incremental experiments, the model is sequentially presented with tasks containing new sets of classes. Each experiment is run with 10 random seeds. As shown in Table 3, when equipped with certain supervised module (DF-CNN), DP-SSCL is able to achieve comparable final testing accuracy to supervised CL using fully labeled data, strictly exceeding that of ORDisCo and DistillMatch. In terms of catastrophic forgetting, we depict backward transfer as a percentage, and so lies in the range $[-100\%, 100\%]$, and the higher value the less forgetting occurs. We can see that DP-SSCL also produces similar forgetting as if fully labeled data is used. This result shows that DP-SSCL is able to capture the properties of continually shifting data and tasks, and generate appropriate pseudo-labels for the lifelong learners.

Table 3: Class-incremental SSCL results.

| | Approach | Final Acc. (%) | | Backward Transfer (%) | |
|---|---|---|---|---|---|
| | | CIFAR-10 | CIFAR-100 | CIFAR-10 | CIFAR-100 |
| *Labeled* | *DF-CNN* | $78.6_{\pm 0.7}$ | $44.8_{\pm 0.6}$ | $6.4_{\pm 2.2}$ | $-3.9_{\pm 0.3}$ |
| *Only* | *DEN* | $60.1_{\pm 2.9}$ | $24.6_{\pm 1.1}$ | $-2.6_{\pm 3.2}$ | $-24.3_{\pm 0.9}$ |
| | *TF* | $78.4_{\pm 0.5}$ | $45.6_{\pm 2.1}$ | $7.8_{\pm 1.3}$ | $-2.1_{\pm 0.4}$ |
| | ORDisCo | 74.8 | not reported | not reported | not reported |
| | DistillMatch | not reported | $24.4_{\pm 0.4}$ | not reported | not reported |
| SSCL | Ours (DF-CNN) | $\mathbf{86.8_{\pm 1.3}}$ | $\mathbf{50.0_{\pm 0.8}}$ | $\mathbf{6.3_{\pm 1.3}}$ | $-7.2_{\pm 0.8}$ |
| | Ours (DEN) | $61.4_{\pm 2.3}$ | $24.1_{\pm 0.6}$ | $-6.4_{\pm 2.5}$ | $-29.6_{\pm 0.7}$ |
| | Ours (TF) | $82.1_{\pm 2.7}$ | $48.9_{\pm 2.1}$ | $1.8_{\pm 2.8}$ | $-7.8_{\pm 2.4}$ |
| *Fully* | *DF-CNN* | $86.8_{\pm 1.4}$ | $52.4_{\pm 1.4}$ | $6.6_{\pm 1.5}$ | $-8.3_{\pm 1.2}$ |
| *Labeled* | *DEN* | $61.4_{\pm 3.7}$ | $23.7_{\pm 0.4}$ | $-6.3_{\pm 3.9}$ | $-31.2_{\pm 0.6}$ |
| | *TF* | $82.5_{\pm 4.6}$ | $51.0_{\pm 1.8}$ | $2.1_{\pm 5.0}$ | $-8.7_{\pm 2.1}$ |

### 5.3.2 SSCL COMPUTATIONAL COST

We analyze DP-SSCL's computational cost required to leverage unlabeled data compared to state-of-the-art SSCL methods, shown in Table 4, where $m$ is the size of one data point, $b$ is the total number of data batches in a task, and $r$ is the number of iterations in CNNL's pseudo-labeling loop. Each scalar parameter in a machine learning model is assumed to take a 4B floating point. We conclude that our DP-SSCL has small overhead in terms of both memory and time for unlabeled data processing, which we elaborate on below.

For memory, CNNL maintains a constant-length queue of data, such that the queue size is proportional to the size of a data point. A typical queue length is 1000 as used in their experiments, with each MNIST data costing 784 B and CIFAR 3072 B. ORDisCo, DistillMatch and DP-SSCL all have a constant $O(1)$ overhead once the architectures are fixed. Nevertheless, in implementation, ORDisCo requires storage of its generator (G) and discriminator (D), which approximately costs 100 MB for parameters, estimated from the last table in (Li et al., 2017) reporting ORDisCo's architecture. DistillMatch requires storage for its OoD detector, which is 254 MB for a ResNet34-based DeConf network (Gao et al., 2021). On the other hand, DP-SSCL utilizes small weak labelers at 0.1MB-level, with 0.083 MB and 0.108 MB for MNIST and CIFAR10/100 benchmarks, respectively. We set a buffer of maximally 25 weak labelers and reallocate the buffer upon each task, so the overhead is $0.083 \times 25 = 2.7$ MB for MNIST and $0.108 \times 25 = 2.75$ MB for CIFAR10/100.

Since the existing SSCL papers do not present timing measures, we compare them using time complexity. Upon every batch of data, the CNNL algorithm repeatedly labels the unlabeled data for an unbounded number of iterations, denoted as $r$, until it is confident of the labels. For ORDisCo, DistillMatch and DP-SSCL, the G + D, OoD detector and weak labelers are trained on all data batches sequentially, so the time overhead is proportional to the number of batches. The conclusion is that DP-SSCL has either lower or the same time complexity to utilize unlabeled data for learning compared to existing SSCL tools.

Table 4: Memory and time overhead to process unlabeled data in different SSCL methods.

|  | ORDisCo | CNNL | DistillMatch | DP-SSCL |
|---|---|---|---|---|
| Source of mem overhead | G and D in GAN | data queue | OoD detector | WLFs |
| Mem complexity | $O(1)$ | $O(m)$ | $O(1)$ | $O(1)$ |
| Mem overhead (MNIST) | not reported | **0.78MB** | not reported | 2.7MB |
| Mem overhead (CIFAR-10) | $\sim$100MB | 3.07MB | not reported | **2.75MB** |
| Mem overhead (CIFAR-100) | not reported | 3.07MB | 254MB | **2.75MB** |
| Source of time overhead | training GAN | repeatedly labeling queue | training OoD detector | training WLFs |
| Time complexity | $O(b)$ | $O(b \times r)$ | $O(b)$ | $O(b)$ |

More experiments are run to show that DP-SSCL improves its performance on larger size of unlabeled data per task, maintains stable learning on increasing number of tasks, and is sensitive to corrupted pseudo-labels. These studies are detailed in Appendix D.

## 6 CONCLUSION

We designed an SSCL framework, namely DP-SSCL, that leverages a DP-based pseudo-labeler and a supervised CL module in cascade manner with a feedback loop. This design allows us to obtain high quality pseudo-labels, shrinking the performance gap between SSCL and supervised CL. Our framework also shows how CL can improve DP by allowing knowledge transfer from previous tasks to improve pseudo-labeling quality based on transferability metrics. Furthermore, the framework is compatible with many existing mature supervised CL approaches, enabling trivial extension from supervised to semi-supervised CL. Our ablation studies show that the framework's performance depend on component selection, and succeeding research shall focus on exploring different component settings more thoroughly. Nevertheless, experiments show DP-SSCL is able to output high lifelong learning accuracy and low forgetting, approaching that of supervised CL on fully labeled data, and outperforming existing SSCL approaches by up to 25% higher final accuracy and lower forgetting, with only 1% of memory overhead at the same time complexity.

## REPRODUCIBILITY STATEMENT

Our experiment code is pushed to an anonymous GitHub repository (https://github.com/dpsscl-anon/DPSSCL). All readers are more than welcomed to replicate our experiments.

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

# Appendices

## A    PROOF OF SNUBA THEORETICAL GUARANTEE IN A LIFELONG SETTING

This section extends Snuba's guaranteed performance, equation 3 in the main paper, to lifelong setting. For completeness, we start by stating proposition:

**Proposition 1** (Snuba's guarantee (Varma & Ré, 2016), adapted to a lifelong setting)**.** *Consider consecutive tasks $\mathcal{Z}^{(1)}, \mathcal{Z}^{(2)}, \ldots$ for which Snuba was used to obtain sets of committed weak labelers $F^{(1)}, F^{(2)}, \ldots$ for all tasks with corresponding empirical accuracies on $X_L^{(i)}$ as a vector $a_L^{(i)}$. For each task, before ensembling, a factor graph-based generative model trains a set of fine-tuned weak labelers from $F^{(i)}$, denoted as $\tilde{F}^{(i)}$ and $|\tilde{F}^{(i)}| = |F^{(i)}|$, that have accuracies on $X_L^{(i)}$ as $\tilde{a}_L^{(i)}$ and on $X_U^{(i)}$ as $\tilde{a}_U^{(i)}$, with $a_L^{(i)}, \tilde{a}_L^{(i)}, \tilde{a}_U^{(i)} \in \mathbb{R}^{|F^{(i)}|}$ and $\tilde{a}_U^{(i)}$ is unknown. We have a measured error $||a_L^{(i)} - \tilde{a}_L^{(i)}||_\infty \le \epsilon^{(i)}$. If each labeler labels a minimum of*

$$d^{(i)} \ge \frac{1}{2(\gamma - \epsilon^{(i)})^2} \log\left(\frac{2|F^{(i)}|^2}{\delta}\right)$$

*data points in $X_L^{(i)}$ with above some given confidence threshold $\nu$ for all iterations, we can guarantee that $||\tilde{a}_U^{(i)} - \tilde{a}_L^{(i)}||_\infty < \gamma$ for all iterations and all tasks with probability $1 - \delta$.*

*Proof.* We know that Snuba ensures the following bound on the labeling performance of an individual task $\mathcal{Z}^{(i)}$ learned in isolation; the following guarantee is a restatement from the original Snuba publication (Varma & Ré, 2016), modified only to include superscripts for the task index $i$:

**Proposition 2** (Snuba's guarantee (Varma & Ré, 2016) for an individual task $\mathcal{Z}^{(i)}$). *Given a set $F^{(i)}$ of committed weak labelers by Snuba for task $\mathcal{Z}^{(i)}$, denote their empirical accuracies on $X_L^{(i)}$ as a vector $a_L^{(i)}$. Before ensembling, a factor graph-based generative model trains a set of fine-tuned weak labelers from $F^{(i)}$, denoted as $\tilde{F}^{(i)}$ and $|\tilde{F}^{(i)}| = |F^{(i)}|$, that have accuracies on $X_L^{(i)}$ as $\tilde{a}_L^{(i)}$ and on $X_U^{(i)}$ as $\tilde{a}_U^{(i)}$, with $a_L^{(i)}, \tilde{a}_L^{(i)}, \tilde{a}_U^{(i)} \in \mathbb{R}^{|F^{(i)}|}$ and $\tilde{a}_U^{(i)}$ is unknown. We have a measured error $||a_L^{(i)} - \tilde{a}_L^{(i)}||_\infty \leq \epsilon$. If each labeler labels a minimum of*

$$d^{(i)} \geq \frac{1}{2(\gamma - \epsilon^{(i)})^2} \log\left(\frac{2|F^{(i)}|^2}{\delta}\right)$$

*data points in $X_L^{(i)}$ with above some given confidence threshold $\nu$ for all iterations, we can guarantee that $||\tilde{a}_U^{(i)} - \tilde{a}_L^{(i)}||_\infty < \gamma$ for all iterations with probability $1 - \delta$.*

See (Varma & Ré, 2016) for the proof of Proposition 2.

We now need to show that this same guarantee holds in a lifelong setting. Assume we have a sequence of $T$ consecutive tasks $\mathcal{Z}^{(1)}, \mathcal{Z}^{(2)}, \ldots, \mathcal{Z}^{(T)}$. Since the first task $\mathcal{Z}^{(1)}$ is learned in isolation to produce the set of weak labelers $F^{(1)}$ using Snuba, by Proposition 2 we know that we have a measured error for the first task of

$$||a_L^{(1)} - \tilde{a}_L^{(1)}||_\infty \leq \epsilon^{(1)}$$

and that

$$||\tilde{a}_U^{(1)} - \tilde{a}_L^{(1)}||_\infty < \gamma$$

for all iterations with probability $1 - \delta$. Let us assume that the bound holds for each $F^{(i)}$ after learning $\mathcal{Z}^{(1)}, \mathcal{Z}^{(2)}, \ldots, \mathcal{Z}^{(T)}$; we will show that this bound also holds for task $\mathcal{Z}^{(T+1)}$.

From tasks $\mathcal{Z}^{(1)}, \ldots, \mathcal{Z}^{(T)}$, Snuba has learned a set of weak labelers $F = \bigcup_{i=1}^{T} F^{(i)}$. When creating the set of weak labelers $F^{(T+1)}$ incrementally, Snuba has two choices at each iteration $j$. Either it can choose to add to $F^{(T+1)}$ an existing weak labeler $f \in F - F^{(T+1)}$ or it can add a previously unused weak labeler $f' \in U - F - F^{(T+1)}$, where $U$ is the pool of all candidate weak labelers. If Snuba chose $f'$ over $f$, then weak labeler $f'$ was assigned a higher score than all other $f \in F$. Similarly, if Snuba chose a particular $f \in F$ to add, then that $f$ was assigned a higher score than all others. Let $\hat{f}_j$ be the weak labeler chosen to add to $F^{(T+1)}$ at iteration $j$, either $f$ or $f'$, which we know has maximum score of all weak labelers in $U - F^{(T+1)}$. Snuba obtains an accuracy on the labeled data set $X_L^{(T+1)}$ of

$$\hat{a}_{L,j}^{(T+1)} = \frac{1}{|\hat{X}_{L,j}^{(T+1)}|} \sum_k \mathbb{1}(y_k^{(T+1)} = \hat{y}_k^{(T+1)}) \ ,$$

where $\hat{X}_{L,j}^{(T+1)} \subseteq X_L^{(T+1)}$ such that $\hat{f}_j$ achieves a confidence greater than or equal to the confidence threshold $\nu$ on each data point in $\hat{X}_L^{(T+1)}$, $y_k^{(T+1)}$ is the true label, and $\hat{y}_k^{(T+1)}$ is the predicted label by the weak labeler $\hat{f}_j$.

(Varma & Ré, 2016) (see their Equation 4) show that the probability of Snuba's failure to maintain $||\tilde{a}_U^{(T+1)} - \tilde{a}_L^{(T+1)}||_\infty < \gamma$ in one iteration is

$$\begin{aligned}
&\Pr[||\tilde{a}_U^{(T+1)} - a_L^{(T+1)}||_\infty + \epsilon \geq \gamma] \\
&\leq 2|F^{(T+1)}| \exp(-2(\gamma - \epsilon)^2 \min(|\hat{X}_{L,1}^{(T+1)}|, \ldots, |\hat{X}_{L,j}^{(T+1)}|)) \ .
\end{aligned} \tag{4}$$

Following (Varma & Ré, 2016) and applying the union bound over the sequence of iterations to bound the probability of failure over all iterations used to acquire $F^{(T+1)}$, we can obtain that

$$\delta \leq 2|F^{(T+1)}|^2 \exp(-2(\gamma - \epsilon^{(T+1)})^2 d^{(T+1)}) \ , \tag{5}$$

where $d^{(T+1)} = \min\left(|\hat{X}_{L,1}^{(T+1)}|, \ldots, |\hat{X}_{L,j}^{(T+1)}|\right)$, and so consequently $||\tilde{a}_U^{(T+1)} - \tilde{a}_L^{(T+1)}||_\infty < \gamma$ similarly holds for $\mathcal{Z}^{(T+1)}$ for all iterations used to obtain $F^{(T+1)}$ with probability $1 - \delta$. By induction, this holds for the entire lifelong sequence.

$\square$

Table 5: Task configurations used in the experiments in Section 5 of the main paper, with one additional committed WLF per iteration, 25 committed WLFs in total at maximum. An interval $[a, b]$ means the hyperparameter is searched within this interval. Notice that 10% of the labeled and unlabeled data are hold-out for validation. Please refer to Figure 4 for the WLF architectures.

| Lifelong Scenario | Task Sequence | # Tasks | Data Split of $n_L^{(i)}, n_U^{(i)}, n_T^{(i)}$ | Weak Labeler Architecture | Learning rate | Batches | Epochs | Bootstrap Size |
|---|---|---|---|---|---|---|---|---|
| Semi-heterogeneous (i.e. 0 vs 1, 0 vs 2, ...) | Binary MNIST Binary CIFAR-10 | 45 45 | 120/11880/2000 400/9600/2000 | Arch A Arch B | [1e-4, 1e-2] [0.8e-3, 1e-3] | [5, 10] 30 | [30, 100] [30, 60] | 30 350 |
| Instance-incremental | 5-way MNIST 10-way CIFAR-10 | 2 1 | 150/29750/5000 2000/30000/10000 | Arch A ($out = 5$) Arch B ($out = 10$) | [0.8e-3, 1.5e-3] 1e-2 | 10 30 | [180, 220] 50 | 50 750 |
| Class-incremental | Binary CIFAR-10 5-way CIFAR-100 | 5 20 | 400/9600/2000 500/2000/500 | Arch B Arch B ($out = 5$) | [0.8e-3, 1e-3] [0.8e-3, 1e-2] | 30 20 | [30, 60] 60, 140 | 350 200 |

## B  DETAILED EXPERIMENTAL SETUP

This section details the experimental setup used in Section 5 of the main paper. Our lifelong learning experiments uses the MNIST, CIFAR-10, and CIFAR-100 benchmark datasets (LeCun & Cortes, 2010; Krizhevsky, 2009).

To transform these benchmarks into the lifelong setting, we created tasks as described in Table 5. For example, to re-create the binary MNIST task sequence, one needs to first arrange the 45 tasks as {0 vs 1, 0 vs 2, ..., 8 vs 9}. Then, split the labeled training data, unlabeled training data, and testing data into 120/11880/2000 per task, ensuring that all data splits have balanced classes. We then hold out 10% of training data for validation. That is, only 108 labeled and 10692 unlabeled data participate in training, with 12 labeled and 1188 used for validation. Finally, perform our procedure described in Section 4 to generate pseudo-labels as well as train a lifelong learner.

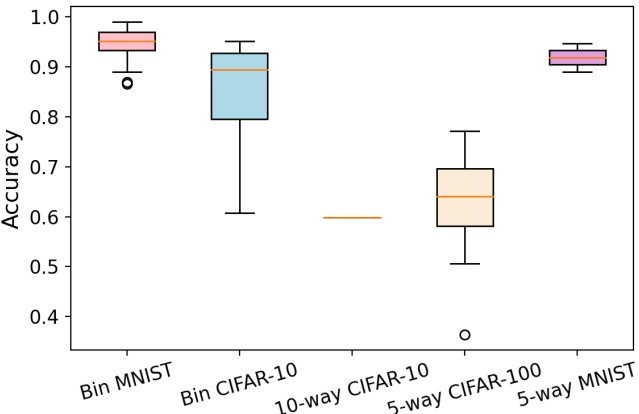

Figure 3: Pseudo-labeling accuracies produced by Snorkel-ensembled labeler.

To train WLFs for data programming, keeps fine-tuning hyperparameters representing the WLF architectures, learning rate, and Snuba configurations in the given search spaces. Since the hyperparameters keep adapting to the current task even throughout a single lifelong sequence, we report these hyperparameters and their constrained search spaces in Table 5 and Figure 4. The search spaces were inspired by various previous works on CNN designs (Lecun et al., 1998; He et al., 2016). Nonetheless, the final architectures are much smaller.

Figure 3 presents the pseudo-labeling accuracy by Snorkel-ensembled pseudo-labeler Ratner et al. (2016a), which is the best labeler according to our ablation studies. The mean accuracies of binary MNIST, binary CIFAR-10, and 5-way MNIST are all around 90%. Although the accuracies of 10-way CIFAR-10 and 5-way CIFAR-100 are $60 - 65\%$ due to difficulty, we can still outperform existing SSCL frameworks as demonstrated in Section 5.3.

We then detail our metric selection for the experiments. The performance of CL can be quantitatively measured by (1) peak per-task performance metrics, (2) final task performance metrics, and (3)

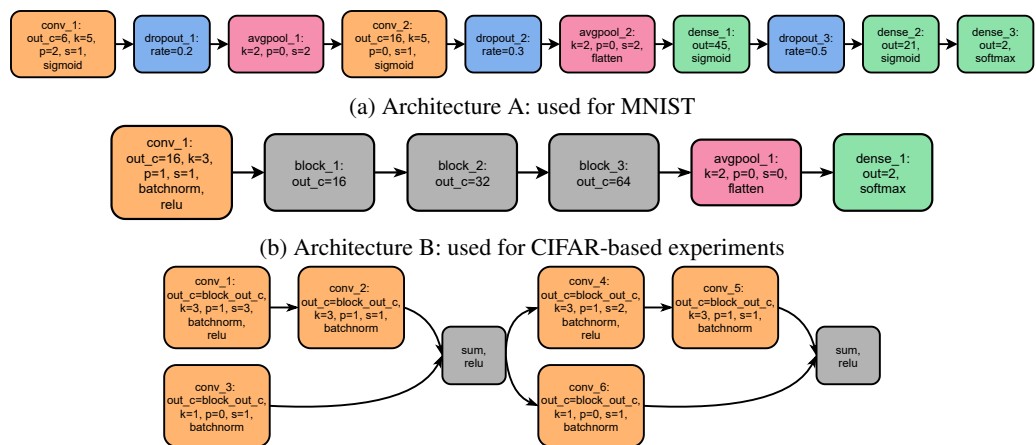

(a) Architecture A: used for MNIST

(b) Architecture B: used for CIFAR-based experiments

(c) Block structure of Architecture B, where $block\_out\_c$ is the $out\_c$ input into the block

Figure 4: WLF model templates at meta initialization of. Notations: $out\_c$: output channels/number of filters, $k$: size (width and height) of a filter, $p$: padding and $s$: stride.

forgetting metrics. After training on task $\mathcal{Z}^{(i)}$, we measure the accuracy $a_j^{(i)}$ of the updated model when evaluated on the testing data of all known tasks $\{\mathcal{Z}^{(j)} : j \leq i\}$. Peak per-task accuracy (Lee et al., 2019) at task $\mathcal{Z}^{(i)}$ measures the average accuracy upon the first encounter of each task: $\tilde{a}_i = \frac{1}{i}\sum_{j=1}^{i} a_j^{(j)}$, and final accuracy (Lopez-Paz & Ranzato, 2017) at task $i$ measures the average performance of the current model on all tasks seen so far: $\bar{a}_i = \frac{1}{i}\sum_{j=1}^{i} a_j^{(i)}$. The retention of knowledge is measured by backward transfer (Ruvolo & Eaton, 2013b; Lopez-Paz & Ranzato, 2017) at task $\mathcal{Z}^{(i)}$ as $bt_i = \frac{1}{i-1}\sum_{j=1}^{i-1} a_j^{(i)} - a_j^{(j)}$. Note that $bt_i \in [-1, 1]$ is the negative of the forgetting metric from (Chaudhry et al., 2019), where positive values indicate improvement and negative values indicate forgetting.

In addition to the lifelong learning metrics, we measure pseudo-labeling accuracy and system overhead of DP-SSCL in terms for memory cost. We also report the proportion of weak labelers using knowledge of earlier tasks and the transferability score to analyze the effect of knowledge sharing on data programming.

## C ADDITIONAL EXPERIMENT ANALYSIS OF SEMI-SUPERVISED CONTINUAL LEARNING

This appendix details and provides additional results that complement Section 5.2 and Section 5.3.1. We start with the ablation studies that are not covered in the main paper.

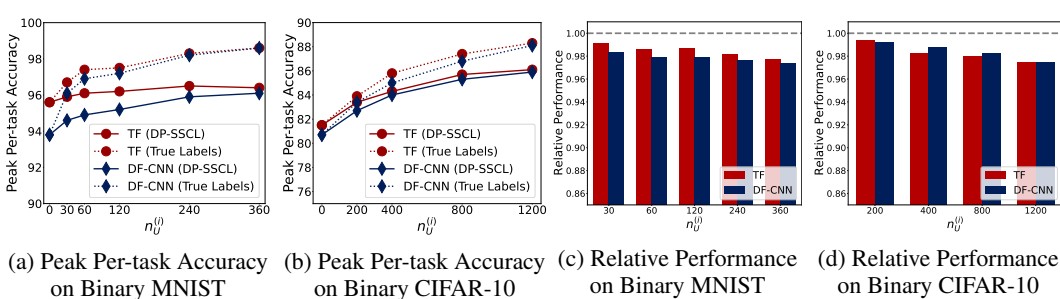

(a) Peak Per-task Accuracy on Binary MNIST  (b) Peak Per-task Accuracy on Binary CIFAR-10  (c) Relative Performance on Binary MNIST  (d) Relative Performance on Binary CIFAR-10

Figure 5: SSCL performance on different sizes of unlabeled data per task.

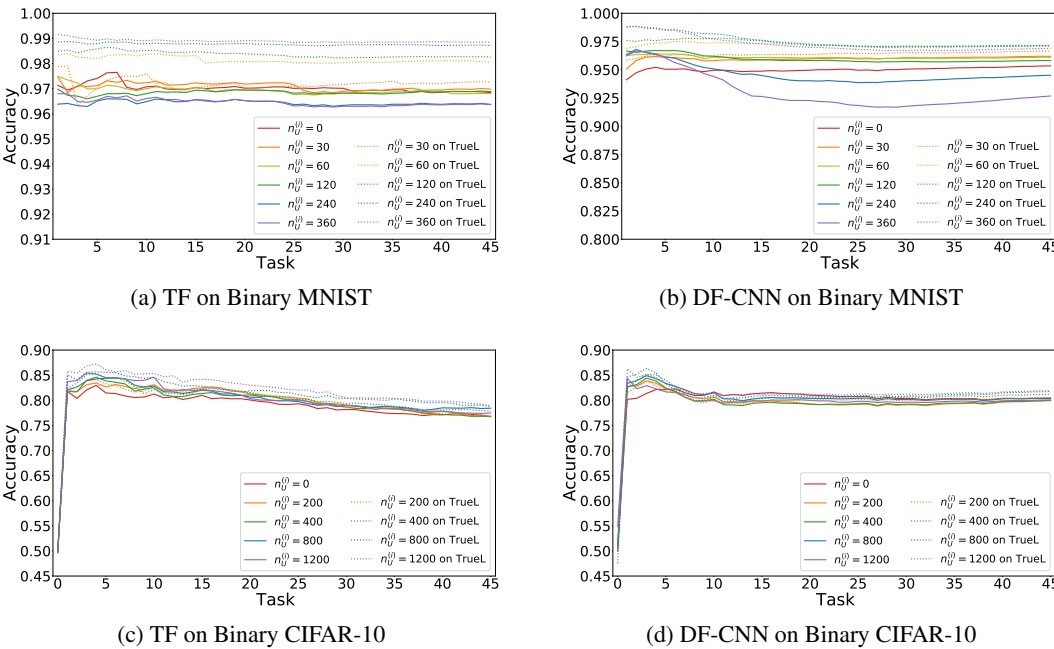

Figure 6: Learning Curve Comparisons. Dotted lines display accuracy of continual learning models trained on unlabeled data $X_U$ with true labels (**TrueL**).

**Sizes of Unlabeled Data** We measure the performance of continual learners with the groundtruth/generated labels on different supervised methods while varying the quantity of unlabeled training data. Table 6 summarizes the empirical results, providing details in addition to Figure 5. Figure 6 visualizes the learning curves of final accuracy in these experiments, showing stable performance of lifelong learners regardless of the size of unlabeled data, number of tasks, and supervised modules.

Peak per-task accuracy of both continual models increases with more DP-SSCL-labeled data. This supports the capability of DP-SSCL to capture the data distribution and assign representative labels. It is well documented that continual models typically have increased negative transfer (i.e., inference or catastrophic forgetting) when trained on more data. However, negative transfer increase is not amplified by using DP-SSCL-generated labels in place of true labels - the continual performance with DP-SSCL-generated pseudo-labels achieves at least $96\%$ of the continual performance when models are trained on the same training data but with true labels instead. Consequently, DP-SSCL maintains stable performance at increasing scale of unlabeled data. In real-world deployment, labeled data has limited availability but unlabeled data is easy to be collected. Therefore, DP-SSCL's scalability supports practical continual learning.

**Number of Tasks** Figure 6 also demonstrates that DP-SSCL preserves relatively stable continual learning performance on increasing number of tasks under different supervised CL modules and task sequences. This result entails that the quality of pseudo-labels generated by DP-SSCL is stable with respect to the number of tasks.

**Pseudo-label Noise Levels** We also examined the effect of introducing noise to DP-SSCL's label generation process: randomly corrupting a portion of the generated labels among the unlabeled data. Figure 7 shows an inverse relationship between manually added pseudo-label noises and lifelong performance. This shows that CL is sensitive to inaccurate pseudo-labels and using more accurate labeling, such as more diverse WLFs on DP, is necessary.

We next detail the ablation study on ensembling methods in Section 5.2.

**Ensembling Methods (More)** We measure the performance of two continual learners, TF (Bulat et al., 2020) and DF-CNN (Lee et al., 2019), on binary MNIST and binary CIFAR-10 experiments

Table 6: Supervised continual learning on binary MNIST (top) and CIFAR-10 (bottom), showing mean $\pm$ standard deviation. Performance is assessed using three metrics, along with accuracy metrics relative to continual models trained on the same data with ground-truth labels instead. * Note: here, backward transfer is depicted as a percentage, and so is scaled by a factor of 100 to lie in range $[-100, 100]$ as compared to the metric definition in Appendix B.

| | $n_U^{(i)}$ | MNIST Performance | | | Relative to True Labels | |
|---|---|---|---|---|---|---|
| | | Per-Task Acc. | Final Acc. | Backward Transfer* | Per-Task | Final Acc. |
| TF | 0 | $95.6_{\pm 0.4}$ | $96.8_{\pm 0.6}$ | $1.9_{\pm 0.1}$ | - | - |
| | 30 | $95.9_{\pm 0.4}$ | $97.0_{\pm 0.4}$ | $1.9_{\pm 0.1}$ | 0.99 | 1.00 |
| | 60 | $96.1_{\pm 0.3}$ | $96.8_{\pm 0.5}$ | $1.8_{\pm 0.1}$ | 0.99 | 0.99 |
| | 120 | $96.2_{\pm 0.2}$ | $96.9_{\pm 0.2}$ | $1.8_{\pm 0.1}$ | 0.99 | 0.99 |
| | 240 | $96.5_{\pm 0.2}$ | $96.4_{\pm 0.2}$ | $1.7_{\pm 0.1}$ | 0.98 | 0.98 |
| | 360 | $96.4_{\pm 0.1}$ | $96.4_{\pm 0.1}$ | $1.8_{\pm 0.1}$ | 0.98 | 0.97 |
| DF-CNN | 0 | $93.8_{\pm 0.4}$ | $95.4_{\pm 0.3}$ | $1.8_{\pm 0.1}$ | - | - |
| | 30 | $94.6_{\pm 0.3}$ | $96.2_{\pm 0.4}$ | $1.5_{\pm 0.2}$ | 0.98 | 1.00 |
| | 60 | $94.9_{\pm 0.2}$ | $96.1_{\pm 0.7}$ | $1.2_{\pm 0.3}$ | 0.98 | 0.99 |
| | 120 | $95.2_{\pm 0.3}$ | $95.8_{\pm 0.9}$ | $0.8_{\pm 0.4}$ | 0.98 | 0.99 |
| | 240 | $95.9_{\pm 0.2}$ | $94.5_{\pm 1.3}$ | $-0.3_{\pm 0.6}$ | 0.98 | 0.97 |
| | 360 | $96.1_{\pm 0.1}$ | $92.7_{\pm 0.7}$ | $-2.0_{\pm 0.7}$ | 0.97 | 0.96 |

| | $n_U^{(i)}$ | CIFAR-10 Performance | | | Relative to True Labels | |
|---|---|---|---|---|---|---|
| | | Per-Task Acc. | Final Acc. | Backward Transfer* | Per-Task | Final Acc. |
| TF | 0 | $81.5_{\pm 0.2}$ | $76.8_{\pm 0.6}$ | $-4.8_{\pm 0.6}$ | - | - |
| | 200 | $83.4_{\pm 0.3}$ | $76.8_{\pm 1.4}$ | $-5.9_{\pm 1.4}$ | 0.99 | 1.00 |
| | 400 | $84.3_{\pm 0.2}$ | $76.7_{\pm 1.3}$ | $-6.6_{\pm 1.0}$ | 0.98 | 0.99 |
| | 800 | $85.7_{\pm 0.1}$ | $78.4_{\pm 0.6}$ | $-5.7_{\pm 0.6}$ | 0.98 | 1.00 |
| | 1200 | $86.1_{\pm 0.2}$ | $77.5_{\pm 1.3}$ | $-7.0_{\pm 1.4}$ | 0.97 | 0.98 |
| DF-CNN | 0 | $80.7_{\pm 0.3}$ | $80.4_{\pm 0.7}$ | $0.2_{\pm 0.7}$ | - | - |
| | 200 | $82.7_{\pm 0.3}$ | $80.0_{\pm 0.5}$ | $-2.0_{\pm 0.6}$ | 0.99 | 0.99 |
| | 400 | $84.0_{\pm 0.3}$ | $80.0_{\pm 0.4}$ | $-2.7_{\pm 0.4}$ | 0.99 | 0.99 |
| | 800 | $85.3_{\pm 0.1}$ | $80.5_{\pm 0.4}$ | $-3.3_{\pm 0.5}$ | 0.98 | 0.98 |
| | 1200 | $85.9_{\pm 0.1}$ | $80.1_{\pm 0.4}$ | $-4.3_{\pm 0.4}$ | 0.98 | 0.98 |

(45 binary classification tasks of pairs of classes such as 0 vs 1, 0 vs 2, ..., 8 vs 9). We ran each MNIST and CIFAR-10 experiment over 10 and 5 random seeds and task sequences, respectively.

Table 7 shows all the peak per-task accuracy, final accuracy and backward transfer with respect to the pseudo-labeling methods and the amount of training data. We see that Snorkel obtains consistently strong performance across continual learning algorithms, data sets, and amounts of unlabeled training data.

Finally, we discuss the learning curves in these experiments. More detailed experiment result statistics are listed in Tables 8, 9, 11 and 10.

**Learning Curve** We include learning curves for the instance-incremental semi-supervised lifelong experiments (Figure 8) and the class-incremental semi-supervised lifelong experiments (Figure 9). All training curves show final accuracy at task $\mathcal{Z}^{(i)}$ as defined in Appendix B, where task $\mathcal{Z}^{(i)}$ is the current task being trained. For easier tasks with high labeling accuracy, such as CIFAR-10, the DP-SSCL-enabled semi-supervised continual learning methods perform similarly to the equivalent

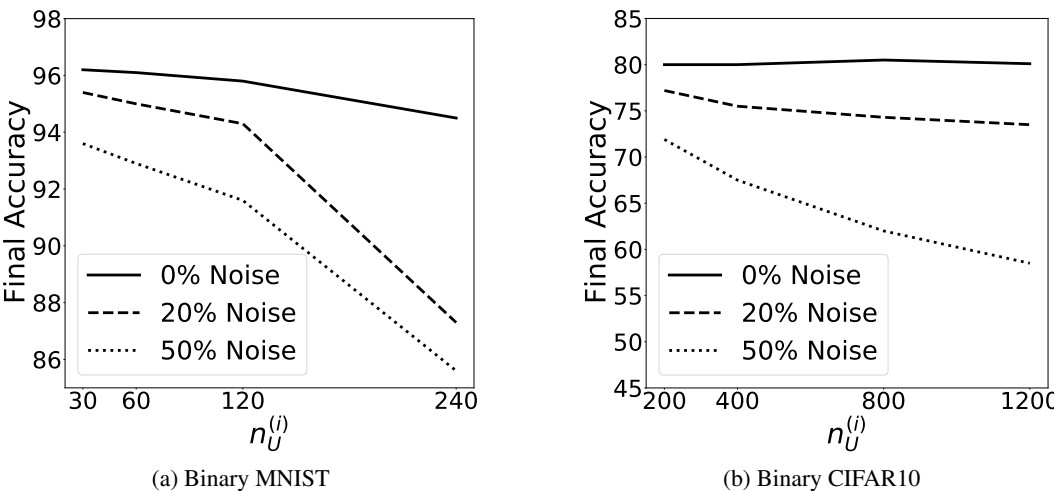

(a) Binary MNIST                                                 (b) Binary CIFAR10

Figure 7: Semi-supervised DF-CNN with three levels of DP-SSCL label noise, showing mean final accuracy.

Table 7: Comparison between pseudo-labeling methods, showing mean final accuracy $\pm$ standard deviation. The training set for each task is a combination of the labeled data (120 and 400 images per task for MNIST and CIFAR-10, respectively) and specified quantity of unlabeled data. Models are trained in a continual learning setting with labels for the unlabeled data generated by one of three weak supervision methods.

| Continual Learner | Labeling Method | MNIST Final Accuracy | | |
| --- | --- | --- | --- | --- |
| | | $n_U^{(i)} = 60$ | $n_U^{(i)} = 120$ | $n_U^{(i)} = 240$ |
| DP-SSCL (TF) | Majority Voting | $96.5 \pm 0.5$ | $89.1 \pm 12.6$ | $94.8 \pm 0.3$ |
| | Repeated Labeling | $89.7 \pm 1.1$ | $84.7 \pm 0.7$ | $77.8 \pm 0.3$ |
| | Snorkel | $\mathbf{97.0} \pm \mathbf{0.5}$ | $\mathbf{97.0} \pm \mathbf{0.2}$ | $\mathbf{96.5} \pm \mathbf{0.2}$ |
| DP-SSCL (DF-CNN) | Majority Voting | $96.0 \pm 0.3$ | $95.6 \pm 0.3$ | $93.3 \pm 1.0$ |
| | Repeated Labeling | $91.8 \pm 0.5$ | $86.1 \pm 1.0$ | $77.4 \pm 0.6$ |
| | Snorkel | $\mathbf{96.5} \pm \mathbf{0.3}$ | $\mathbf{96.4} \pm \mathbf{0.3}$ | $\mathbf{95.2} \pm \mathbf{0.8}$ |
| Continual Learner | Labeling Method | CIFAR10 Final Accuracy | | |
| | | $n_U^{(i)} = 400$ | $n_U^{(i)} = 800$ | $n_U^{(i)} = 1200$ |
| DP-SSCL (TF) | Majority Voting | $\mathbf{77.4} \pm \mathbf{0.8}$ | $\mathbf{76.0} \pm \mathbf{0.5}$ | $\mathbf{76.5} \pm \mathbf{1.0}$ |
| | Repeated Labeling | $76.3 \pm 0.9$ | $74.9 \pm 0.9$ | $73.9 \pm 0.9$ |
| | Snorkel | $\mathbf{77.3} \pm \mathbf{0.6}$ | $\mathbf{76.1} \pm \mathbf{0.8}$ | $75.6 \pm 1.1$ |
| DP-SSCL (DF-CNN) | Majority Voting | $78.1 \pm 0.7$ | $77.0 \pm 1.1$ | $76.0 \pm 0.9$ |
| | Repeated Labeling | $77.0 \pm 0.4$ | $75.2 \pm 0.7$ | $74.4 \pm 0.9$ |
| | Snorkel | $\mathbf{79.1} \pm \mathbf{0.9}$ | $\mathbf{78.6} \pm \mathbf{0.7}$ | $\mathbf{78.5} \pm \mathbf{0.8}$ |

supervised fully-labeled task sequence. For a further breakdown of results, we include task-specific performance measures for all of the semi-supervised lifelong experiments in Tables 8–11.

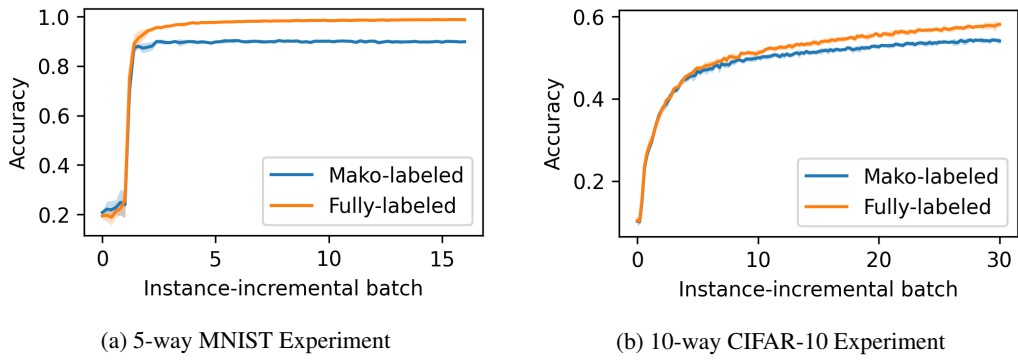

(a) 5-way MNIST Experiment

(b) 10-way CIFAR-10 Experiment

Figure 8: Instance-incremental semi-supervised vs fully supervised learning curve comparisons

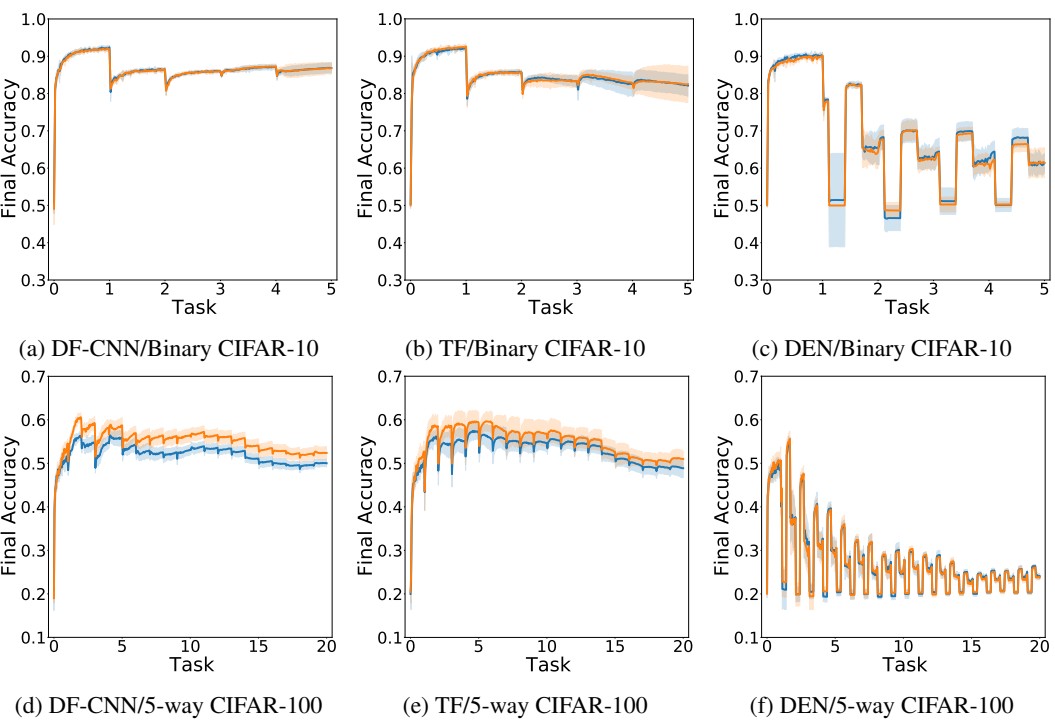

(a) DF-CNN/Binary CIFAR-10

(b) TF/Binary CIFAR-10

(c) DEN/Binary CIFAR-10

(d) DF-CNN/5-way CIFAR-100

(e) TF/5-way CIFAR-100

(f) DEN/5-way CIFAR-100

Figure 9: Class-incremental semi-supervised (blue) vs fully supervised (orange) learning curve comparisons, showing standard deviation with shaded regions.

Table 8: Class-incremental Binary CIFAR-10 experiment results broken down by task, showing mean ± standard deviation

| Approach | Task | DP-SSCL (Semi-supervised) | | Fully-labeled (Supervised) | |
|---|---|---|---|---|---|
| | | Final Acc. | Back. Transfer (%) | Final Acc. | Back. Transfer (%) |
| DF-CNN | 0 | $88.1 \pm 1.4$ | $-4.3 \pm 1.2$ | $86.7 \pm 5.3$ | $-5.1 \pm 5.2$ |
| | 1 | $75.9 \pm 4.8$ | $-4.6 \pm 5.2$ | $76.6 \pm 1.7$ | $-4.4 \pm 1.9$ |
| | 2 | $84.4 \pm 2.0$ | $-1.3 \pm 1.8$ | $84.7 \pm 1.1$ | $-1.1 \pm 1.2$ |
| | 3 | $93.2 \pm 0.5$ | $-0.2 \pm 0.7$ | $93.5 \pm 0.7$ | $-0.1 \pm 0.5$ |
| | 4 | $92.5 \pm 1.1$ | $0.0 \pm 0.0$ | $92.6 \pm 0.6$ | $0.0 \pm 0.0$ |
| TF | 0 | $80.5 \pm 7.2$ | $-11.7 \pm 7.0$ | $79.0 \pm 11.2$ | $-13.5 \pm 11.1$ |
| | 1 | $67.8 \pm 6.7$ | $-12.5 \pm 6.7$ | $71.0 \pm 5.0$ | $-9.1 \pm 5.2$ |
| | 2 | $78.1 \pm 8.8$ | $-6.7 \pm 9.1$ | $77.4 \pm 9.4$ | $-8.0 \pm 9.7$ |
| | 3 | $91.3 \pm 3.5$ | $-2.7 \pm 3.9$ | $92.4 \pm 1.4$ | $-1.8 \pm 1.2$ |
| | 4 | $92.9 \pm 0.5$ | $0.0 \pm 0.0$ | $92.8 \pm 1.0$ | $0.0 \pm 0.0$ |
| DEN | 0 | $50.4 \pm 0.7$ | $-40.1 \pm 0.9$ | $51.7 \pm 4.3$ | $-38.2 \pm 4.5$ |
| | 1 | $50.2 \pm 3.3$ | $-26.1 \pm 3.4$ | $50.8 \pm 9.4$ | $-24.9 \pm 9.2$ |
| | 2 | $58.1 \pm 9.7$ | $-23.7 \pm 9.5$ | $54.5 \pm 8.7$ | $-27.6 \pm 8.6$ |
| | 3 | $60.7 \pm 7.5$ | $-29.8 \pm 7.5$ | $59.9 \pm 5.4$ | $-30.7 \pm 5.7$ |
| | 4 | $87.4 \pm 3.9$ | $0.0 \pm 0.0$ | $90.0 \pm 0.6$ | $0.0 \pm 0.0$ |

Table 9: Class-incremental 5-way CIFAR-100 experimental results with the DF-CNN, broken down by task, showing mean ± standard deviation

| Approach | Task | DP-SSCL (Semi-supervised) | | Fully-labeled (Supervised) | |
|---|---|---|---|---|---|
| | | Final Acc. | Back. Transfer (%) | Final Acc. | Back. Transfer (%) |
| DF-CNN | 0 | $33.9 \pm 5.4$ | $-17.0 \pm 4.5$ | $31.8 \pm 6.0$ | $-21.6 \pm 6.4$ |
| | 1 | $35.1 \pm 6.5$ | $-29.4 \pm 6.5$ | $33.8 \pm 6.3$ | $-35.8 \pm 6.5$ |
| | 2 | $32.8 \pm 4.0$ | $-27.6 \pm 4.6$ | $32.8 \pm 5.7$ | $-32.5 \pm 6.1$ |
| | 3 | $41.6 \pm 6.6$ | $-23.4 \pm 6.4$ | $38.8 \pm 5.9$ | $-30.2 \pm 6.5$ |
| | 4 | $37.5 \pm 7.5$ | $-33.5 \pm 7.7$ | $42.0 \pm 7.2$ | $-31.9 \pm 7.0$ |
| | 5 | $45.8 \pm 5.4$ | $-13.9 \pm 5.6$ | $50.7 \pm 7.6$ | $-13.7 \pm 7.2$ |
| | 6 | $49.3 \pm 5.1$ | $-10.3 \pm 5.4$ | $51.5 \pm 4.5$ | $-12.7 \pm 4.3$ |
| | 7 | $46.7 \pm 3.5$ | $-14.3 \pm 3.2$ | $53.6 \pm 3.7$ | $-14.3 \pm 2.9$ |
| | 8 | $57.2 \pm 3.1$ | $-6.0 \pm 2.9$ | $58.2 \pm 5.1$ | $-8.3 \pm 4.8$ |
| | 9 | $64.2 \pm 2.7$ | $-4.9 \pm 2.4$ | $65.9 \pm 2.3$ | $-5.5 \pm 3.2$ |
| | 10 | $70.1 \pm 3.1$ | $-4.0 \pm 4.1$ | $71.1 \pm 3.0$ | $-5.2 \pm 2.0$ |
| | 11 | $53.0 \pm 3.5$ | $-3.7 \pm 3.7$ | $57.7 \pm 2.5$ | $-2.4 \pm 2.3$ |
| | 12 | $61.7 \pm 1.7$ | $-2.9 \pm 1.8$ | $64.2 \pm 1.7$ | $-2.4 \pm 2.5$ |
| | 13 | $54.1 \pm 1.8$ | $-1.7 \pm 2.1$ | $55.4 \pm 1.8$ | $-4.9 \pm 2.7$ |
| | 14 | $33.0 \pm 1.7$ | $-1.2 \pm 1.1$ | $36.7 \pm 1.9$ | $-1.2 \pm 2.1$ |
| | 15 | $50.0 \pm 1.3$ | $0.0 \pm 1.6$ | $52.7 \pm 2.5$ | $-1.4 \pm 2.5$ |
| | 16 | $48.1 \pm 2.2$ | $-0.2 \pm 2.5$ | $50.0 \pm 2.3$ | $-1.0 \pm 2.4$ |
| | 17 | $53.1 \pm 1.2$ | $0.1 \pm 1.6$ | $56.1 \pm 1.6$ | $0.5 \pm 2.8$ |
| | 18 | $62.7 \pm 2.1$ | $0.1 \pm 2.3$ | $67.7 \pm 1.8$ | $0.0 \pm 1.3$ |
| | 19 | $70.5 \pm 1.8$ | $0.0 \pm 0.0$ | $77.3 \pm 1.9$ | $0.0 \pm 0.0$ |

Table 10: Class-incremental 5-way CIFAR-100 experimental results with TF, broken down by task, showing mean ± standard deviation

| Approach | Task | DP-SSCL (Semi-supervised) | | Fully-labeled (Supervised) | |
|---|---|---|---|---|---|
| | | Final Acc. | Back. Transfer (%) | Final Acc. | Back. Transfer (%) |
| | 0 | $36.9 \pm 6.7$ | $-13.5 \pm 7.5$ | $38.8 \pm 9.1$ | $-13.2 \pm 9.2$ |
| | 1 | $43.4 \pm 4.8$ | $-19.8 \pm 5.6$ | $49.9 \pm 9.9$ | $-16.5 \pm 9.8$ |
| | 2 | $36.2 \pm 7.0$ | $-20.5 \pm 6.8$ | $37.5 \pm 7.0$ | $-23.9 \pm 7.6$ |
| | 3 | $56.4 \pm 5.2$ | $-8.7 \pm 5.2$ | $54.0 \pm 5.5$ | $-15.0 \pm 4.9$ |
| | 4 | $40.6 \pm 13.2$ | $-29.4 \pm 13.4$ | $42.4 \pm 10.7$ | $-29.3 \pm 11.7$ |
| | 5 | $51.0 \pm 7.4$ | $-9.3 \pm 7.5$ | $50.9 \pm 13.6$ | $-16.1 \pm 13.5$ |
| | 6 | $46.3 \pm 7.1$ | $-14.4 \pm 7.6$ | $52.6 \pm 5.0$ | $-10.0 \pm 5.8$ |
| | 7 | $48.8 \pm 9.3$ | $-11.1 \pm 8.7$ | $50.8 \pm 7.4$ | $-13.2 \pm 5.9$ |
| | 8 | $49.6 \pm 8.7$ | $-14.4 \pm 7.3$ | $47.0 \pm 8.5$ | $-17.3 \pm 8.0$ |
| | 9 | $60.4 \pm 7.1$ | $-7.9 \pm 7.0$ | $58.5 \pm 6.0$ | $-12.1 \pm 6.3$ |
| TF | 10 | $57.5 \pm 11.4$ | $-16.8 \pm 11.8$ | $56.9 \pm 14.0$ | $-21.3 \pm 12.9$ |
| | 11 | $48.3 \pm 8.3$ | $-8.8 \pm 7.5$ | $46.9 \pm 6.0$ | $-12.7 \pm 6.9$ |
| | 12 | $53.6 \pm 5.2$ | $-7.3 \pm 5.2$ | $57.6 \pm 3.8$ | $-6.3 \pm 4.5$ |
| | 13 | $48.4 \pm 5.5$ | $-6.4 \pm 5.9$ | $52.9 \pm 8.2$ | $-9.8 \pm 9.0$ |
| | 14 | $31.0 \pm 2.0$ | $-2.9 \pm 2.0$ | $31.6 \pm 2.9$ | $-2.4 \pm 2.4$ |
| | 15 | $41.5 \pm 6.0$ | $-8.0 \pm 6.1$ | $48.1 \pm 2.8$ | $-3.7 \pm 2.4$ |
| | 16 | $45.1 \pm 3.0$ | $-2.1 \pm 2.7$ | $45.2 \pm 5.2$ | $-3.3 \pm 4.3$ |
| | 17 | $50.0 \pm 4.3$ | $-2.5 \pm 5.2$ | $54.5 \pm 2.4$ | $-1.2 \pm 2.2$ |
| | 18 | $62.8 \pm 2.7$ | $-1.7 \pm 2.6$ | $67.8 \pm 2.2$ | $-3.1 \pm 2.7$ |
| | 19 | $69.6 \pm 2.2$ | $0.0 \pm 0.0$ | $76.4 \pm 2.7$ | $0.0 \pm 0.0$ |

Table 11: Class-incremental 5-way CIFAR-100 experimental results with the DEN, broken down by task, showing mean ± standard deviation

| Approach | Task | DP-SSCL (Semi-supervised) | | Fully-labeled (Supervised) | |
|---|---|---|---|---|---|
| | | Final Acc. | Back. Transfer (%) | Final Acc. | Back. Transfer (%) |
| | 0 | $21.1 \pm 2.9$ | $-28.2 \pm 3.5$ | $19.6 \pm 1.1$ | $-31.8 \pm 1.4$ |
| | 1 | $18.6 \pm 2.4$ | $-41.6 \pm 2.3$ | $22.7 \pm 2.3$ | $-38.4 \pm 2.8$ |
| | 2 | $20.3 \pm 2.2$ | $-35.6 \pm 1.6$ | $21.0 \pm 2.4$ | $-37.7 \pm 3.5$ |
| | 3 | $20.1 \pm 1.6$ | $-40.9 \pm 2.8$ | $20.5 \pm 3.0$ | $-43.0 \pm 3.3$ |
| | 4 | $24.5 \pm 4.7$ | $-42.3 \pm 4.8$ | $20.1 \pm 1.5$ | $-47.4 \pm 3.1$ |
| | 5 | $21.9 \pm 1.4$ | $-34.7 \pm 1.5$ | $21.8 \pm 2.5$ | $-37.4 \pm 2.3$ |
| | 6 | $19.3 \pm 1.0$ | $-38.8 \pm 1.5$ | $19.6 \pm 2.7$ | $-41.1 \pm 3.5$ |
| | 7 | $22.4 \pm 2.5$ | $-35.7 \pm 2.2$ | $21.8 \pm 2.3$ | $-39.6 \pm 2.9$ |
| | 8 | $22.0 \pm 1.3$ | $-34.7 \pm 1.7$ | $21.8 \pm 1.3$ | $-31.0 \pm 1.6$ |
| | 9 | $24.0 \pm 2.6$ | $-42.4 \pm 2.2$ | $22.0 \pm 2.7$ | $-44.1 \pm 3.4$ |
| DEN | 10 | $22.7 \pm 1.8$ | $-48.3 \pm 2.3$ | $23.1 \pm 3.1$ | $-50.4 \pm 2.4$ |
| | 11 | $22.9 \pm 1.9$ | $-31.4 \pm 2.9$ | $22.8 \pm 2.5$ | $-33.3 \pm 2.3$ |
| | 12 | $21.9 \pm 2.1$ | $-38.1 \pm 2.5$ | $22.3 \pm 1.4$ | $-38.5 \pm 1.4$ |
| | 13 | $24.4 \pm 3.0$ | $-28.8 \pm 3.1$ | $24.4 \pm 3.3$ | $-31.9 \pm 3.4$ |
| | 14 | $20.9 \pm 1.5$ | $-12.6 \pm 2.3$ | $20.2 \pm 1.1$ | $-11.7 \pm 2.0$ |
| | 15 | $22.8 \pm 3.0$ | $-26.0 \pm 2.6$ | $20.9 \pm 1.6$ | $-29.6 \pm 1.8$ |
| | 16 | $21.2 \pm 0.8$ | $-26.3 \pm 1.6$ | $21.1 \pm 2.1$ | $-27.2 \pm 2.4$ |
| | 17 | $22.2 \pm 1.6$ | $-31.8 \pm 3.2$ | $21.6 \pm 1.7$ | $-33.6 \pm 2.2$ |
| | 18 | $23.1 \pm 2.1$ | $-38.0 \pm 2.6$ | $23.0 \pm 2.0$ | $-39.8 \pm 2.8$ |
| | 19 | $65.2 \pm 3.5$ | $0.0 \pm 0.0$ | $64.1 \pm 5.2$ | $0.0 \pm 0.0$ |

# D  ADDITIONAL EXPERIMENT ANALYSIS OF DP-SSCL WITH KNOWLEDGE TRANSFER IN WEAK LABELERS

This appendix details and provides additional results that complement Section 5.2 with respect to the transfer of weak labeling functions across tasks. As described in Section 4.2, we can utilize the previously trained weak labelers for the current task if the current task has enough similarity with some earlier tasks. The key component is the suitability score in Algorithm 1 determining which earlier tasks or which earlier WLFs are sufficiently related to the current task.

The main paper showed experimental results using the LEEP transferability score as the suitability score. In this appendix, we experiment with other suitability scores. Specifically we compare the following similarity measures:

- **the LEEP transferability score**, as used in the original paper. LEEP provides task-level information on the similarity of earlier tasks to the current one.

- **LEEP + Snuba score**. Instead of only relying on task transferability, this measure incorporates Snuba's s-score (Equation 1) to measure how earlier WLFs perform on the current task prior to fine-tuning. Specifically, we computed it as $s_{WLF}(w) = \alpha_L \mathrm{LEEP}(\mathcal{M}, X_L^{(i)}, y_L^{(i)}) + \alpha_F \mathrm{F1}(w, X_L^{(i)}, y_L^{(i)}) + \alpha_J \mathrm{Jaccard}(w, X_L^{(i)}, y_L^{(i)})$ where $w \in \mathcal{W}$ is an individual WLF, $\mathcal{M}$ is a CL model and $\{\alpha_L, \alpha_F, \alpha_J\}$ are weights of the sum. In this experiment, all terms were weighted identically ($\alpha_L = \alpha_F = \alpha_J = 1/3$), and the LEEP score was projected to be in the range $[0, 1]$ for compatibility with the other terms via the exponential function $((-\inf, 0) \to (0, 1))$.

- **the OTCE transferability score**. OTCE Tan et al. (2021) measures the similarity of earlier tasks to the current one with respect to the distance between probability distributions.

- **OTCE + Snuba score**. Similar to LEEP + Snuba score, this measure incorporates the individual performance of earlier WLFs as well as task-wise relationships. We computed this score as $s_{WLF}(w) = \alpha_O \mathrm{OTCE}(X_L^{(i)}, y_L^{(i)}, X_L^{(1:i-1)}, y_L^{(1:i-1)}) + \alpha_F \mathrm{F1}(w, X_L^{(i)}, y_L^{(i)}) + \alpha_J \mathrm{Jaccard}(w, X_L^{(i)}, y_L^{(i)})$ where $w \in \mathcal{W}$ is an individual WLF, $X_L^{(1:i-1)}$ and $y_L^{(1:i-1)}$ are labeled data of earlier tasks, and $\{\alpha_L, \alpha_F, \alpha_J\}$ are weights of the sum. In this experiment, the OTCE score was projected to be in the range $[0, 1]$ by linear transformation, and all terms were weighted identically ($\alpha_L = \alpha_F = \alpha_J = 1/3$).

We evaluated the different suitability scores in our approach on the CIFAR-10 dataset, using 45 binary classification tasks formed by all pairs of two image classes (see Table 5). As in the experiments from Section 5.2, DF-CNN is used as the continual learner. For both experiments of $(n_L^{(i)}, n_U^{(i)}) = (400, 200)$ and $(n_L^{(i)}, n_U^{(i)}) = (50, 400)$, transferring WLFs makes the continual learner achieve equal or better peak per-task accuracy and final accuracy compared to the performance of DP-SSCL without WLF transfer. This experiment shows that this result is robust to the choice of suitability score.

| CIFAR-10 Experiment | Approach | Performance | |
| --- | --- | --- | --- |
| | | Per-Task Acc. | Final Acc. |
| | No Transfer | $81.9_{\pm 0.1}$ | $77.4_{\pm 1.0}$ |
| $n_L^{(i)} = 400$ | LEEP ($\phi = -0.5, \rho = 0.6$) | $82.1_{\pm 0.2}$ | $78.9_{\pm 0.8}$ |
| $n_U^{(i)} = 200$ | LEEP + Snuba Score ($\phi = 0.7, \rho = 0.6$) | $82.1_{\pm 0.2}$ | $78.8_{\pm 0.3}$ |
| | OTCE ($\phi = 0.8, \rho = 0.6$) | $82.3_{\pm 0.3}$ | $78.3_{\pm 0.5}$ |
| | OTCE + Snuba Score ($\phi = 0.75, \rho = 0.6$) | $82.5_{\pm 0.2}$ | $79.2_{\pm 0.4}$ |
| $n_L^{(i)} = 50$ | No Transfer | $75.3_{\pm 0.5}$ | $72.3_{\pm 0.7}$ |
| $n_U^{(i)} = 400$ | LEEP ($\phi = -0.3, \rho = 0.4$) | $75.5_{\pm 0.3}$ | $72.4_{\pm 0.7}$ |
| | LEEP + Snuba Score ($\phi = 0.7, \rho = 0.6$) | $75.8_{\pm 0.5}$ | $72.4_{\pm 0.4}$ |

Table 12: SSCL performance when initializing WLFs via knowledge transfer

