# OpenReview forum: "Bridging the Gap between Semi-supervised and Supervised Continual Learning via Data Programming"
_ICLR.cc/2023/Conference — Submitted to ICLR 2023_

### Official Review · Reviewer_SCJ5 · 2022-10-25

**Confidence:** 3
**Correctness:** 1
**Technical Novelty And Significance:** 2
**Empirical Novelty And Significance:** Not applicable
**Recommendation:** 5

**Clarity, Quality, Novelty And Reproducibility:**

The paper is well written with minor linguistic errors. The structure is well organized and the content is easy to follow. The semi-supervised continual learning idea has been presented clearly with minimum fuzziness.

There have been a number of innovations especially the DP-SSCL framework itself including the evaluation and experimental protocol that has been followed (mainly to compare against available techniques in the literature). A number of novel ideas can be traced in the DP-SSCL pipeline  steps:  (1) Meta (parameter) initialization, (2) Weak Labeling Function (WLF) initialization via knowledge transfer, (3) WLF training and pruning, and (4) Pseudo-labeler ensembling and continual model update.

As the code has been shared, it is expected the results can be reproduced although there has been some experimental setup problem identified as explained in the weakness section.




**Strength And Weaknesses:**

Strengths and weakness:
Strengths: The paper is well written clearly covering the problem definition, the scope, the experimental and evaluation protocol, the evaluation metrics and the setup to compare with existing techniques in the field. It provides a set of thorough experiments that highlight the effectiveness of approach and the methodology. The reported results look not only better than existing approaches but sometimes close to the completely supervised setup (although tested on simple and small datasets). Also, DP-SSCL reports less memory and time complexity when compared to other existing approaches. In addition, the proposed framework is compatible with other Continuous Learning (CL) approaches so it can be easily evaluated and compared with others.

Weakness: As reported in the paper, the Ensembling Methods and the WLF Transfer are two very important steps of the DP-SSCL framework. It looks the corresponding parameters of these two steps (section 5.3) were learned/chosen using the same data that has been used to report and compare (with the competitors) DP-SSCL performance in section 5.4. These underlying data should have been exclusive for steps 5.3 and 5.4, and therefore the results of section 5.4 may not be valid. It is suggested that authors redo these experiments by setting a disjoint set of data for these two steps. Same comments are applicable for the setup in Appendix B.


**Summary Of The Paper:**

In this paper, the authors have proposed a  Data Programming (DP) based Semi-supervised Continual Learning (SSCL)  framework which they call as DP-SSCL. Data programming (DP) is a known technique and its objective is to assign pseudo labels to unlabelled data points as a part of the semi supervised learning (SSL) process. The main idea of this paper is to first label unlabelled data points using DP and  then use those as additional (to already available labeled data) labels to further consolidate the model.

The proposed DP-SSCL framework follow the following steps (in the cascading order) including a feedback loop: (1) Meta (parameter) initialization, (2) Weak Labeling Function (WLF) initialization via knowledge transfer, (3) WLF training and pruning, and (4) Pseudo-labeler ensembling and continual model update. The proposed framework has been tested on  MNIST and CIFAR benchmark datasets and  compared against (a) some existing SSCL methods such as CNNL, ORDisCo, DistillMatch, and (b) Fully supervised Continual Learning with labeled data only.  A set of the performance metrics: 1) peak per-task performance, (2) final task performance, and (3) forgetting scores have been also researched and used in model comparisons.  Reported result is found to be promising.

**Summary Of The Review:**

I have gone through the paper more than once including the appendices. Overall, the idea is quite sound, well articulated through the document. The experiment is thorough and the reported results are encouraging.

There has been a major flaw identified in the experiment section 5 as some of the model comparisons were done using parameters that were selected based on the same data (see the weakness section). This might have over judged/scored the model performance.

The authors are suggested to redo their experiments with a proper setup for a valid comparison. I think this work has some value if this problem can be resolved.

---

> ### Author Response · Authors · 2022-11-15
> **Response to reviewer SCJ5**
>
> Thank you for your thoughtful comments.
>
> Regarding parameter selection in the experiments:
> “The corresponding parameters of these two steps (section 5.3) were learned/chosen using the same data that has been used to report and compare (with the competitors) DP-SSCL performance in section 5.4. These underlying data should have been exclusive for steps 5.3 and 5.4, and therefore the results of section 5.4 may not be valid. It is suggested that authors redo these experiments by setting a disjoint set of data for these two steps.”
>
> We thank the reviewer for this insight – what you suggested is exactly how we conducted the experiment. We apologize for the lack of clarity on this, and have revised the paper to make it clearer. Please notice that Section 5.3 is now re-indexed to 5.2 and 5.4 is now 5.3 in the revision.
>
> For ablation studies in 5.2, we report the results on a validation data split, and based on that we choose pipeline components for the main experiments in 5.3, which is evaluated on a separate testing data split. We do not select components by outcomes on the same data. (Note the difference between Table 6 and Table 7). In our revised paper, this procedure is further clarified in Section 5.1 and 5.2, and the generation of validation data split is specified in Appendix B.

---

### Official Review · Reviewer_HUWP · 2022-10-25

**Confidence:** 4
**Correctness:** 4
**Technical Novelty And Significance:** 3
**Empirical Novelty And Significance:** 3
**Recommendation:** 8

**Clarity, Quality, Novelty And Reproducibility:**

The paper is clearly written and is of high quality. The authors have released their code and have used public datasets which ensures reproducibility. As mentioned above, the paper is not extremely novel, but solves an important problem and does it well using practical techniques.

**Strength And Weaknesses:**

*Strengths*
1. The paper solves an important problem and has impressive results, beating previous reported state-of-the-art methods.
2. I like the fact that the authors have taken time to analyze the memory and time complexity of their methods with baselines.
3. The authors have a theoretical justification of the quality of labeling function proposed by the mechanisms used by Snuba under the continual learning settings.

*Weaknesses*
1. A quick google scholar search brings up a number of papers on semi-supervised continual learning [1, 2, 3, 4] which seem to not have been covered by the authors in either the literature review or as baselines.
2. Comparison on Tiny-imagenet: Previous methods such as DistillMatch also compared their performance on Tiny-Imagenet. I believe that the paper would benefit from experiments on more datasets.
3. I believe that the paper does not have extremely novel methodology. Having said that, I believe that the paper solves an important problem and does it well using practical methods.

References:
[1] Brahma, Dhanajit, Vinay Kumar Verma, and Piyush Rai. "Hypernetworks for Continual Semi-Supervised Learning." arXiv preprint arXiv:2110.01856 (2021).
[2] Luo, Yan, et al. "Learning to Predict Gradients for Semi-Supervised Continual Learning." arXiv preprint arXiv:2201.09196 (2022).
[3] Boschini, Matteo, et al. "Continual semi-supervised learning through contrastive interpolation consistency." Pattern Recognition Letters 162 (2022): 9-14.
[4] Ho, Stella, et al. "Semi-supervised Continual Learning with Meta Self-training." Proceedings of the 31st ACM International Conference on Information & Knowledge Management. 2022.

**Summary Of The Paper:**

The paper proposes a semi-supervised continual learning technique which leverages data programming to probabilistically label unlabeled data. The paper leverages methods proposed in Snuba to automatically generate labeling functions for new tasks.

**Summary Of The Review:**

I think the paper is strong and would benefit from comparison with some related work, some more experiments. I would also like the authors to highlight the novelty of their approach.

---

> ### Author Response · Authors · 2022-11-15
> **Response to reviewer HUWP**
>
> We appreciate the reviewer’s acknowledgement of our work.
>
> 1. Regarding citing additional SSCL papers: “A number of papers on semi-supervised continual learning [1, 2, 3, 4] which seem to not have been covered by the authors in either the literature review or as baselines”
>
> We thank the reviewer for pointing out these literature we missed. We have now cited and briefly discussed these four papers in our revised Section 2.2.
>
> 2. Regarding running additional experiments on TinyImageNet: "Previous methods such as DistillMatch also compared their performance on Tiny-Imagenet. I believe that the paper would benefit from experiments on more datasets.”
>
> We believe this is a great idea. We are running these experiments and we will post the results as soon as we have them. We are hoping to finish them before the revision deadline (November 18) and we will update you before it.
>
> We have also corrected grammatical errors we found in the revised version, as pointed out.
>
> References: [1] Brahma, Dhanajit, Vinay Kumar Verma, and Piyush Rai. "Hypernetworks for Continual Semi-Supervised Learning." arXiv preprint arXiv:2110.01856 (2021). [2] Luo, Yan, et al. "Learning to Predict Gradients for Semi-Supervised Continual Learning." arXiv preprint arXiv:2201.09196 (2022). [3] Boschini, Matteo, et al. "Continual semi-supervised learning through contrastive interpolation consistency." Pattern Recognition Letters 162 (2022): 9-14. [4] Ho, Stella, et al. "Semi-supervised Continual Learning with Meta Self-training." Proceedings of the 31st ACM International Conference on Information & Knowledge Management. 2022.

---

> > ### Author Response · Authors · 2022-11-19
> > **Update on TinyImageNet experiments**
> >
> > Unfortunately, we are unable to finish the TinyImageNet experiments before the revision deadline (November 18), but the experiments are still running. We will update the results in the comment section once they are out. Thank you for understanding.

---

### Official Review · Reviewer_5ri5 · 2022-10-25

**Confidence:** 3
**Correctness:** 3
**Technical Novelty And Significance:** 2
**Empirical Novelty And Significance:** 3
**Recommendation:** 5

**Clarity, Quality, Novelty And Reproducibility:**

- Clarity: The paper is overall clearly written.

- Novelty and Quality: Applying data programming to CL is novel on its own, but there are limited additional technical contributions. More comprehensive experiments and deeper analysis can make the paper better.

- Reproducibility: The authors released the code anonymously.


**Strength And Weaknesses:**

Strengths:
- The idea of applying data programming to continual learning is new. The connection and synergy between the two are natural, i.e., similar tasks in continual learning could share similar WLFs, making data programming specifically suitable for CL.
- By adopting data programming, the underlying CL methods do not have to be modified. This makes the proposed pipeline flexible and able to enjoy improvements as new CL methods are developed.

Weaknesses:
- While the combination of CL and data programming is novel, the technical contribution appears to be more limited. “Transferability” could be one aspect where the authors could more carefully look into to improve over previously used metrics like LEEP. However, this seems to be not studied in depth.
- Since the combination of DP and SSCL is new, Sec 5.3 should be one main focus of the paper, but the current set of experiments shown appears to be somewhat weak. For example, only one Transferability metric, LEEP, is tested in the experiments.
- The entire DP-SSCL pipeline requires many different hyperparemeters to be set, including $\phi$ and $\rho$. How are the hyperparameters selected in the experiments?
- Table 3 lacks one baseline that readers would be interested in seeing, i.e., CL with only the labeled data. It is also encouraged to show the dataset statistics, i.e., size of the labeled set, unlabeled set, test set, etc.
- In the introduction, it is mentioned that applying DP to CL can be more robust to distribution shifts. However, it is not clear to me why this is the case, and I also didn’t find any experimental results supporting this.


**Summary Of The Paper:**

The paper proposes to apply Data Programming to semi-supervised Continual Learning (SSCL). The core idea is to first generate weak labeling functions (WLF) with existing tools like Snuba, where the labeling functions are then used to pseudo-label the unlabeled data points that are eventually all fed into training the downstream continual learning models. Experimental results show that this pipeline leads to improved performance over existing SSCL methods and performs more closely to the fully-supervised continual learning methods.

**Summary Of The Review:**

I like the idea of applying Data Programming to SSCL, the current empirical results also show that this is a promising approach. However, I believe more comprehensive analysis on the proposed DP-SSCL pipeline can make the results stronger, for example, answering questions like what are the important aspects on selecting the transferability metric.

---

> ### Author Response · Authors · 2022-11-15
> **Response to reviewer 5ri5**
>
> Thank you for your detailed comments.
>
> We address the next two excellent points together:
>
> 1. “Weakness 1: Transferability could be one aspect where the authors could more carefully look into to improve over previously used metrics like LEEP. However, this seems to be not studied in depth.”
>
> 2. “Weakness 2: The current set of experiments shown appears to be somewhat weak. For example, only one Transferability metric, LEEP, is tested in the experiments.”
>
> These are great points; we’ve now added a new experimental section (in Appendix D) to explore using different measures as the suitability score in Algorithm 1. As you mention, the LEEP transferability score (as explored in our original experiments) is just one example that could be used as the suitability criterion. We’ve now added a comparison to another (incorporating F1 score and Jaccard distance) in Appendix D, and are currently working to add additional experimental results with other metrics (e.g., OTCE). We’ll add those additional results to Appendix D once they’re available.
>
> 3. “Weakness 3: The entire DP-SSCL pipeline requires many different hyperparameters to be set, including $\phi$ and $\rho$. How are the hyperparameters selected in the experiments?”
>
> For the current task, we set the hyperparameters based on the held-out validation data sets for all tasks encountered up to that current task. Specifically, when training on task $i$, $\phi$ and $\rho$ are based on the validation data for tasks $1, \ldots, i$. This ensures that the hyperparameters are only based upon known tasks, without knowledge of future tasks. We have clarified this in Section 5.2 (previously 5.3) and we have explained the train/validation/test split in Appendix B.
>
> 4. “Weakness 4: Table 3 lacks one baseline that readers would be interested in seeing, i.e., CL with only the labeled data. It is also encouraged to show the dataset statistics, i.e., size of the labeled set, unlabeled set, test set, etc.”
>
> Great idea. We are currently running this experiment, and will add a follow-up post once these results are available. For the dataset statistics, we had included those in Appendix B.
>
> 5. “Weakness 5: In the introduction, it is mentioned that applying DP to CL can be more robust to distribution shifts. However, it is not clear to me why this is the case, and I also didn’t find any experimental results supporting this.”
>
> There was a lack of clarity on our part – thanks for catching this. We were referring to the natural change in distribution between different tasks encountered during continual learning, and we see that the term “distribution shift” is imprecise and suggests a different evaluation setting. We have revised the end of Section 2.2 to clarify that we are referring to changes in data that arise from task changes, which should also make it clear that our statement is indeed encompassed by the experiments in Section 5.3.1 (previously 5.4.1), which show our method outperforms other SSCL methods that do not use non-data-programming methods for pseudo-labeling.

---

> > ### Author Response · Authors · 2022-11-19
> > **Update on Table 3 and Table 12 (November 18 2022)**
> >
> > We have finished running additional experiments and updated the results in our revised paper.
> >
> > 1. For weakness 4, we ran additional baseline experiments using labeled data only, and the result is added to the beginning of Table 3 in our latest revision. Based on these results, our DP-SSCL framework is able to improve up to 8% final accuracy by leveraging additional unlabeled data.
> >
> > 2. For weakness 1 and 2, we finished experiments on additional transferability metrics (OTCE and OTCE + Snuba score) and the result is added to Table 12 in Appendix D.

---

### Official Review · Reviewer_LGhA · 2022-10-27

**Confidence:** 2
**Clarity, Quality, Novelty And Reproducibility:** These are all seem fine.
**Correctness:** 3
**Technical Novelty And Significance:** 3
**Empirical Novelty And Significance:** 3
**Recommendation:** 5

**Strength And Weaknesses:**

Pros:
* They consider a more realistic continual learning with only a few labeled data, and adopt a novel technique, data programming, to annotate these unlabeled data.

* From the perspective of engineering, they greatly reduce the computational cost.

Cons:
* This paper seems to lack a type of important baseline. They can incorporate the existing semi-supervised learning training strategies into the continual learning methods to overcome the weak supervision problem. They need to justify the advantages of DP-SSCL over such straightforward solutions.

* I have a doubt about the meta initialization process. They use the task sequence as the prior knowledge to initialize hyperparameters and model architectures. However, for continual learning, could we obtain the full task sequence at once? If yes, why do not we use the full task sequence to perform standard supervised learning.

* For more severe scenarios, we could not obtain qualified WLFs. Under such a setting, whether DP-SSCL is still useful?

**Summary Of The Paper:**

This paper proposes to use data programming to address the semi-supervised continuous learning, achieving the performance close to the fully supervised continual learning.

**Summary Of The Review:**

Please refer to Strength And Weaknesses.

---

> ### Author Response · Authors · 2022-11-15
> **Response to reviewer LGhA**
>
> Thank you for your insightful comments. We address your concerns as follows.
>
> 1. “This paper seems to lack a type of important baseline. They can incorporate the existing semi-supervised learning training strategies into the continual learning methods to overcome the weak supervision problem. They need to justify the advantages of DP-SSCL over such straightforward solutions.”
>
> We agree that comparison to existing, relevant baselines is important.  Thank you for raising this point; however, we’re not entirely clear on the other baseline you’re proposing.  As discussed in Section 2.2, there are very few semi-supervised continual learning methods. Section 5.3.1 (previous 5.4.1) includes a comparison to a number of other semi-supervised CL baselines from the literature, each of which integrate a different semi-supervised labeling approach–e.g. ORDisCo uses GAN-based pseudo-labeling and CNNL uses incremental pseudo-labeling of new data using its current model–and extends them to continual learning settings. Our work does the same with data programming, and thus we compare to the previous semi-supervised approaches that have successfully been applied to continual learning in prior work.
>
> Are you suggesting that we take those same strategies and incorporate them directly into the base continual learners that we use in the evaluation (e.g., DF-CNN) without using data programming (and in such a way that isn’t covered by these other methods)? If so, constructing such new algorithms are likely to be substantial contributions on their own, worthy of their own individual papers. Or, do you instead have some other straightforward way of combining these that we’ve overlooked?  Would you please clarify, and let us know if we misinterpreted this concern?
>
> 2. “I have a doubt about the meta initialization process….for continual learning, could we obtain the full task sequence at once? If yes, why do not we use the full task sequence to perform standard supervised learning.?”
>
> During the meta-initialization process, we are providing only general information for the types of tasks the learner is facing, such as general model architectures and hyperparameter tuning strategies. Critically, we are NOT providing information during the meta-initialization on the tasks themselves, their data, the sequence of tasks, their distribution, or their relationships. In the continual learning setting, the task sequence is not available a priori, and so information about the tasks is only obtained as the tasks arrive consecutively. We have clarified this in a revision to the first few sentences of Section 4.1.
>
> 3. “For more severe scenarios, we could not obtain qualified WLFs. Under such a setting, whether DP-SSCL is still useful?”
>
> Data programming only requires WLFs to be better than random guessing, which is a low requirement in general for learning agents. Even in complex scenarios, data programming has been shown to be quite effective [1, 2]. Consequently, we anticipate that DP-SSCL would be effective wherever data programming in general is effective. We have mentioned this in a revision to Section 2.3.
>
> References: [1] Pugh, Sydney, et al. "High-Confidence Data Programming for Evaluating Suppression of Physiological Alarms." 2021 IEEE/ACM Conference on Connected Health: Applications, Systems and Engineering Technologies (CHASE). IEEE, 2021. [2] Cirillo, Davide, and Alfonso Valencia. "Big data analytics for personalized medicine." Current opinion in biotechnology 58 (2019): 161-167.

---

### Author Response · Authors · 2022-11-15
**Response to Reviewers**

We thank the reviewers for all of their thoughtful comments. To address this feedback, we have uploaded a new version of the paper, with changes shown in blue for ease of identification, along with a new appendix (Appendix D).  We also reply to the individual concerns raised by each reviewer below, and hope to continue discussion with reviewers to clarify any points that we misinterpreted, or that the reviewers feel are still not adequately addressed.

The major changes include:
1. Additional experiments (currently in progress) using TinyImageNet, as suggested,
2. An additional section (Appendix D) with experiments varying the transferability/similarity metric, as suggested, and
3. Clarifications to address concerns in the reviews, including clarification hyperparameter tuning in our experimental procedure.
4. To fit within the page limit, we moved the detailed explanation of evaluation metrics from Section 5.2 to Appendix B, and deleted the old 5.2. Therefore, some subsections in Section 5 are re-indexed (5.4 becomes 5.3 and 5.3 becomes 5.2).

---

### Author Response · Authors · 2022-12-01
**We are eager to discuss with the reviewers**

Dear reviewers of our paper, thanks again for your insightful comments. We would like to let you know that we have responded to your concerns and modified the paper correspondingly in our revision. It will be much appreciated if you can check whether your concerns are addressed properly, let us know if anything else is further needed, and update the recommendation score accordingly. Thank you in advance.

---

### Decision · Program_Chairs · 2023-01-20

**Decision:**

Reject

**Justification For Why Not Higher Score:**

Most reviewers expressed reservations about various aspects of empirical evaluation part of the paper.

**Justification For Why Not Lower Score:**

n/a

**Metareview: Summary, Strengths And Weaknesses:**

This paper has been assessed by four knowledgeable reviewers. It proposed an application of data programming to continual learning scenarios, which is novel, but the technical and experimental infrastructure surrounding the concept, as it stands right now, does not appear sufficient. This limits the perceived novelty of the work as well as it diminishes strength of the technical contribution. Three of the reviewers rated this paper below the threshold of ICLR acceptance while on voted to accept it as is. This work has some merit but it is not ready yet to be published at ICLR.